



**Biological production in two contrasted regions of the Mediterranean Sea during the**
**oligotrophic period: An estimate based on the diel cycle of optical properties measured**
**by BGC-Argo profiling floats**
Marie Barbieux[1], Julia Uitz[1], Alexandre Mignot[2], Collin Roesler[3], Hervé Claustre[1], Bernard
Gentili[1], Vincent Taillandier[1], Fabrizio D'Ortenzio[1], Hubert Loisel[4], Antoine Poteau[1],
Edouard Leymarie[1], Christophe Penkerc'h[1], Catherine Schmechtig[5], Annick Bricaud[1]
[1]CNRS and Sorbonne Université, Laboratoire d'Océanographie de Villefranche, LOV, 06230 Villefranche-sur-
Mer, France
[2]Mercator Océan, 31520 Ramonville-Saint-Agne, France
[3]Bowdoin College, Earth and Oceanographic Science, Brunswick, Maine 04011, USA
[4]Université Littoral Côte d'Opale, Université Lille, CNRS, Laboratoire d'Océanologie et de Géosciences, 59000
Lille, France
[5]OSU Ecce Terra, UMS 3455, CNRS and Sorbonne Université, Paris 6, 4 place Jussieu, 75252 Paris CEDEX 05,
France
Correspondence to: J. Uitz (julia.uitz@imev-mer.fr)



**Abstract**
This study assesses marine biological production of organic carbon based on the diel
variability of bio-optical properties monitored by two BioGeoChemical-Argo (BGC-Argo)
floats. Experiments were conducted in two distinct Mediterranean systems, the Northwestern
Ligurian Sea and the Central Ionian Sea during summer months. We derived particulate
organic carbon (POC) stock and gross community production integrated within the surface,
euphotic and subsurface chlorophyll maximum (SCM) layers, using an existing approach
applied to diel cycle measurements of the particulate beam attenuation ($c_p$) and backscattering
($b_{bp}$) coefficients. The diel cycle of $c_p$ provided a robust proxy for quantifying biological
production in both systems; that of $b_{bp}$ was comparatively less robust. Derived primary
production estimates vary by a factor of 2 depending upon the choice of the bio-optical
relationship that converts the measured optical coefficient to POC, which is thus a critical step
to constrain. Our results indicate a substantial, yet variable, contribution to the water column
production of the SCM layer (16–42%). In the Ligurian Sea, the SCM is a seasonal feature
that behaves as a subsurface biomass maximum (SBM) with the ability to respond to episodic
abiotic forcing by increasing production. In contrast, in the Ionian Sea, the SCM is permanent,
induced by phytoplankton photoacclimation and contributes moderately to water column
production. These results emphasize the strong potential for transmissometers deployed on
BGC-Argo profiling floats to quantify non-intrusively *in situ* biological production of organic
carbon in the water column of stratified oligotrophic systems with recurring or permanent
SCMs, which are widespread features in the global ocean.



## 1   Introduction

Primary production is an essential component of the global ocean carbon cycle (Field et

al. 1998). As a major driver of the biological carbon pump, this biogeochemical process plays
a critical role in the regulation of the Earth climate (e.g. Sarmiento & Siegenthaler 1992;
Falkowski 2012). Hence, quantifying primary production in the ocean stands as a major
challenge in the context of climate change. The balance between gross primary production
and community respiration in the ocean determines the trophic status of marine systems, i.e.
whether the system acts as a source or a sink of carbon (Williams 1993). This balance
depends on the considered region and varies substantially according to spatial and temporal
scales (Geider et al. 1997; Duarte & Agusti 1998; del Giorgio & Duarte 2002). It is therefore
necessary to develop capabilities not only for assessing primary production on a global scale,
but also for characterizing and quantifying the biogeochemical functioning of marine
ecosystems at smaller spatial and temporal scales (Serret et al. 1999; González et al. 2001 &

2002).

Traditionally, primary production measurements are based on *in situ* or *in vitro*

incubation experiments (i.e. on board the ship, under controlled conditions) coupled with
isotopic carbon analysis (Nielsen 1952; Fitzwater et al. 1982; Dandonneau 1993; Barber &
Hitling 2002) or measurements of oxygen concentration (Williams & Jenkinson 1982;
Williams & Purdie 1991). These methods involve seawater sampling during field campaigns,
sample manipulation and subsequent laboratory analyses, which are both time consuming and
require strong technical expertise. As a result, the availability of field primary production
measurements is relatively limited in terms of spatial and temporal coverage, which hinders
the possibility of extrapolation to other systems or to larger space and time scales for
modeling purposes.



Bio-optical primary production models coupled with ocean color satellite imagery
represent another approach for obtaining primary production estimates (Morel 1991;
Longhurst et al. 1995; Antoine et al. 1996; Behrenfeld et al. 2002). Such models are
extremely valuable for assessing primary production with a large spatial coverage and over a
broad range of temporal scales (Sathyendranath et al. 1995; Uitz et al. 2010; Chavez et al.
2011). Yet, most of these models suffer from several sources of uncertainties that generate
potential errors in the production estimates (e.g. Sarmiento et al. 2004; Saba et al. 2010; Saba
et al. 2011). Sources of uncertainties include, in particular, the extrapolation of the satellite
chlorophyll product, which is weighted to the upper portion of the euphotic zone, to the
entirety of the productive region of the water column not sensed remotely. In addition, the *in*
*situ*-based parameterization of phytoplankton photophysiology tends to lack robustness when
applied to large (regional or global) scales and over seasonal to interannual time scales.
Diel cycles observed in bio-optical properties provide a less-empirical and more
mechanistic approach to assess biological production. In a seminal paper published in 1989,
Siegel et al. observed the *in situ* diurnal variability of the particulate beam attenuation
coefficient ($c_p$) and used it as a surrogate for the diurnal variations in the abundance of
biogenic particles and associated production in the oligotrophic North Pacific Ocean. Several
studies subsequently pursued the investigation of the diurnal variability of marine bio-optical
properties as a means for determining non-intrusively *in situ* biological production (e.g.
Stramska & Dickey 1992; Durand and Olson 1996; Claustre et al. 1999; Claustre 2008;
Gernez et al. 2011; White et al. 2017; Briggs et al. 2018).
Among this large body of literature, Claustre et al. (2008) carried further the principle
of the Siegel et al. (1989) approach for application to the South Pacific Subtropical Ocean.
Based upon the generally observed relationship between the $c_p$ coefficient and the stock of
particulate organic carbon, POC (e.g. Stramski et al. 1999; Garner et al. 2006), Claustre et al.



(2008) assumed that diel variations in $c_p$ reflect diel variations in POC. Thus, the observed
daytime increase and nighttime decrease in $c_p$-derived POC are used to estimate gross
community production, community losses and, assuming equivalent day and night losses, net
community production (NCP). Because the $c_p$ coefficient is not specific to phytoplankton but
includes the POC contribution of both autotrophic and heterotrophic particles, the $c_p$-based
method yields estimate of community production.
Two studies (Kheireddine & Antoine 2014; Barnes & Antoine 2014) also attempted to
extend this approach to the particulate backscattering coefficient ($b_{bp}$). The application opens
up opportunities for assessing community production from geostationary ocean color satellite
observations, from which a nearly continuous daytime $b_{bp}$ coefficient can be retrieved. Both
studies focused on surface data obtained from moored observations from the Ligurian Sea
(Northwestern Mediterranean) and found weak results for the diel $b_{bp}$ cycle, calling for further
investigations.
The optics-based approach has proven to be particularly relevant for appraising
particulate biological production in stratified oligotrophic systems such as subtropical gyres
(e.g. Siegel et al. 1998; Claustre et al. 2008; White et al. 2017). Interestingly, in such systems,
the biological production of organic carbon is difficult to quantify and potentially
underestimated by $^{14}C$ incubation methods (Juranek & Quay 2005; Quay et al. 2010). This
might be attributed to an inadequacy of traditional measurement methods for apprehending
the spatial and temporal heterogeneity of biological production that may exhibit local or
episodic events (Karl et al. 2003; Williams et al. 2004; McGillicuddy 2016). Moreover, in
stratified oligotrophic systems, the vertical distribution of phytoplankton is frequently
characterized by the presence of a deep chlorophyll maximum (DCM), also referred as
subsurface chlorophyll maximum (SCM; e.g. Cullen 1982; Hense & Beckmann 2008; Cullen
2014; Mignot et al., 2014). SCMs are not necessarily resolved by *in situ* discrete sampling and



cannot be observed from ocean color satellites that are limited to the surface ocean. They are
typically attributed to the photoacclimation of phytoplankton cells to low light conditions
(Kiefer et al. 1976; Cullen 1982; Fennel & Boss 2003; Letelier et al., 2004; Dubinsky &
Stambler 2009). Yet, SCMs resulting from an actual increase in phytoplankton (carbon)
biomass, and so referred to as subsurface biomass maximum (SBM), have also been observed
episodically and/or seasonally in oligotrophic regions of the global ocean (Beckmann &
Hense 2007; Mignot et al. 2014; Barbieux et al. 2019; Cornec et al. 2021). Considering the
large (45%) surface areas covered by stratified oligotrophic regions in the global ocean
(McClain et al. 2004), improving the quantification of biological production of organic carbon
and characterizing the contribution of SCMs to the water-column production in such regions
are critical. For this purpose, *in situ* diel-resolved measurements with high spatio-temporal
resolution in the entire water column represent an intriguing opportunity.

In this study, we exploit summertime observations acquired by two BioGeoChemical-

Argo (BGC-Argo) profiling floats deployed in contrasted systems of the Mediterranean Sea.
This offers a unique opportunity for pursuing the exploration of the bio-optical diel cycle-
based approach to biological production in oligotrophic environments. One of the two BGC-
Argo floats was deployed in the Ligurian Sea in the vicinity of the BOUSSOLE fixed
mooring (BOUée pour l'acquiSition d'une Série Optique à Long termE; Antoine et al. 2008).
This area is representative of a seasonally stratified oligotrophic system, with a potentially
productive SCM (e.g. Mignot et al. 2014; Barbieux et al. 2019) that follows a recurrent spring
bloom. The second float was deployed in the Ionian Sea (Central Mediterranean) as part of
the PEACETIME (ProcEss studies at the Air-sEa Interface after dust deposition in the
MEditerranean sea) project (Guieu et al. 2020). The Ionian Sea is a nearly permanent
oligotrophic system (e.g. Lavigne et al. 2015) with an SCM induced mostly by
photoacclimation (e.g. Mignot et al. 2014; Barbieux et al. 2019).



The BGC-Argo profiling floats used in this study measured, among a suite of physical
and biogeochemical properties, the $c_p$ and $b_{bp}$ coefficients and were both programmed to
sample the entire water column at a high temporal resolution (4 vertical profiles per 24h), in
order to monitor the diel variations of the bio-optical properties. We applied, for the first time,
a modified version of the method of Claustre et al. (2008) to the diel $c_p$ and $b_{bp}$ measurements
acquired by the BGC-Argo floats to derive community production. Using this dataset, we (1)
assess the relevance of the diel cycle-based method for estimating biological production of
organic carbon in the considered regions and discuss the applicability of the method to $b_{bp}$, in
addition to $c_p$; (2) investigate the regional and vertical variability of the production estimates
with a focus on the SCM layer in relation to the biological and abiotic context; (3) discuss the
relative contribution of the SCM layer to the water-column community production.
**2    Data and methods**
**2.1    Study region**
The Mediterranean Sea provides a unique environment for investigating the
biogeochemical functioning of oligotrophic systems that exhibit either a seasonal or
permanent SCM. The Mediterranean is a deep ocean basin characterized by a West-to-East
gradient in nutrients and chlorophyll *a* concentration (e.g. Dugdale & Wilkerson 1988;
Bethoux et al. 1992; Antoine et al. 1995; Bosc et al. 2004; D'Ortenzio & D'Alcalà 2009)
associated with a deepening of the SCM (Lavigne et al. 2012; Barbieux et al. 2019). The
Ionian Sea in the eastern Mediterranean is defined as permanently oligotrophic, with the SCM
settled at depth over the whole year. This system represents the oligotrophic end-member type
of SCM (Barbieux et al. 2019), much like the subtropical South Pacific Ocean Gyre. By
contrast, the Ligurian Sea in the western Mediterranean is seasonally productive as akin to a
temperate system (e.g. Casotti et al. 2003; Marty & Chiavérini 2010; Siokou-Frangou et al.



2010; Lavigne et al. 2015). The mixed layer deepens significantly during the winter period,
inducing seasonal renewal of nutrients in the surface layer that supports the spring bloom
(Marty et al. 2002; Lavigne et al. 2013; Pasqueron de Fommervault et al. 2015; Mayot et al.
2016). After the seasonal bloom, the SBM intensifies throughout the summer and into early
fall. This system represents the temperate end-member type of SCM.

We deployed BGC-Argo floats programmed for "multi-profile" sampling in each of

these two regions (Fig. 1). The Ligurian Sea float (hereafter noted fLig, WMO: 6901776),
was deployed in the vicinity of the BOUSSOLE fixed mooring (7°54'E, 43°22'N) during one
of the monthly cruises of the BOUSSOLE program (Antoine et al. 2008). We used the fLig
float measurements acquired during the time period May 24 to September 13, 2014. The
Ionian Sea float (hereafter noted fIon, WMO: 6902828) was deployed as part of the
PEACETIME project (Guieu et al. 2020). We used the fIon float measurements acquired
during the time period May 28 to September 11, 2017. Thus, although collected in different
years, the data sets arise from similar seasonal contexts.
**2.2    BGC-Argo multi-profiling floats and data processing**

The BGC-Argo floats used in this study are "PROVOR CTS-4" (nke Instrumentation,

Inc.). They were both equipped with the following sensors and derived data products: (1) a
CTD sensor for depth, temperature and salinity; (2) a "remA" combo sensor that couples a
SAtlantic OCR-504 (for downwelling irradiance at three wavelengths in addition to
photosynthetic available radiation, PAR) and a WET Labs ECO Puck Triplet (for both
chlorophyll *a* (excitation/emission wavelengths of 470 nm/695 nm) and colored dissolved
organic matter (CDOM; 370 nm/460 nm) fluorescence, and particulate backscattering
coefficient at 700 nm); and (3) a WET Labs C-Rover (for particulate beam attenuation
coefficient at 660 nm, 25-cm pathlength). Data were collected along water column profiles
from 1000 m up to the surface with a vertical resolution of 10 m between 1000 and 250 m, 1





m between 250 and 10 m, and 0.2 m between 10 m and the surface. First, the BGC-Argo raw
counts were converted into geophysical units by applying factory calibration. Second, we
applied corrections following the BGC-Argo QC procedures (Schmechtig et al. 2015, 2016;
Organelli et al. 2017).

Factory-calibrated chlorophyll fluorescence requires additional corrections for

determining the chlorophyll *a* concentration (Chl). Values collected during daylight hours
were corrected for non-photochemical quenching following Xing et al. (2012). A global
analysis of factory-calibrated chlorophyll fluorescence measured with WET Labs ECO
sensors relative to concurrent chlorophyll *a* concentrations, determined by High Performance
Liquid Chromatography (HPLC), yielded a global overestimate bias of 2 (Roesler et al. 2017),
with statistically significant regional biases varying between 0.5 and 6. In the Mediterranean
Sea, the regional variations of the fluorescence-to-Chl ratio are known to be very small
(Taillandier et al. 2018), hence the bias correction factor of 2 was applied to BGC-Argo
fluorescence data from both the Ligurian and Ionian regions, consistently with the processing
performed at the Coriolis Data Center.

For the particulate backscattering coefficient ($b_{bp}$), we followed the BGC-Argo

calibration and quality control procedure of Schmechtig et al. (2016). The backscattering
coefficient at 700 nm ($m^{-1}$) is retrieved following Eq. (1):
$$b_{bp}(700) = 2\,\pi\,\chi\,[(\beta b_{bp} - Dark b_{bp}) \times Scale b_{bp} - \beta sw] \qquad (1)$$
where $\chi = 1.076$ is the empirical weighting function that converts particulate volume
scattering function at 124° to total backscattering coefficient (Sullivan et al. 2013); $\beta b_{bp}$ is
the raw observations from the backscattering meter (digital counts); $Dark b_{bp}$ (digital counts)
and $Scale b_{bp}$ ($m^{-1}$ $sr^{-1}$ $count^{-1}$) are the calibration coefficients provided by the manufacturer;
and $\beta sw$ is the contribution to the Volume Scattering Function (VSF) by the pure seawater at



the 700 nm measurement wavelength that is a function of temperature and salinity (Zhang et
al. 2009).

The calibration procedure applied to the particulate beam attenuation coefficient ($c_p$) is

similar to that described in Mignot et al. (2014). The beam transmission, T (%), is
transformed into the beam attenuation coefficient, c (m$^{-1}$), using the relationship:
$c = -\frac{1}{x} \ln \frac{T}{100}$                                                  (2)
where x is the transmissometer pathlength (25 cm). The beam attenuation coefficient c is the
sum of the absorption and scattering by seawater and its particulate and dissolved
constituents. At 660 nm, the contribution of CDOM ($c_{cdom}$) can be considered negligible in
oligotrophic waters because, although its absorption in the blue is comparable to that of
particulate material (Organelli et al. 2014), the $c_{cdom}$ spectrum decays exponentially towards
near zero in the red (Bricaud et al. 1981), and because it is comprised of dissolved molecules
and colloids, its scattering is negligible (Boss and Zaneveld 2003). Meanwhile $c_w(660)$ for
pure water is constant and removed in the application of the factory calibration; effects due to
dissolved salt are accounted for according to Zhang et al. (2009). Hence, at a wavelength of
660 nm, the particle beam attenuation coefficient, $c_p$ (m$^{-1}$), is retrieved by subtracting the
seawater contribution to c. The biofouling-induced increasing signal observed in clear deep
waters and resulting in a drift in $c_p$ values with time, is corrected as follows. For each profile,
a median $c_p$ value, used as an "offset", is computed from the $c_p$ values acquired between 300
m and the maximum sampled depth, and subtracted from the entire profile.

Using the solar noon Photosynthetically Available Radiation (PAR) measurements, we

computed the euphotic layer depth ($Z_{eu}$) as the depth at which the PAR is reduced to 1% of its
value just below the surface (Gordon & McCluney 1975) and the penetration depth ($Z_{pd}$; also
known as the e-folding depth or first attenuation depth) as $Z_{eu}$ / 4.6. We define the surface





layer from 0 m to $Z_{pd}$. We also define the SCM layer as in Barbieux et al. (2019), whereby a
Gaussian model is fit to each Chl vertical profile in order to determine the depth interval of
the full width half maximum of the SCM. Finally, the Mixed Layer Depth (MLD) is derived
from the float CTD data as the depth at which the potential density difference relative to the
surface reference value is 0.03 kg m$^{-3}$ (de Boyer Montégut et al. 2004).

Unlike the majority of BGC-Argo floats that collect profile measurements every 10

days, the two platforms used in this study sampled the water column 4 profiles per day, albeit
with slightly different regimes (Fig. 2). The fLig float cycle commences with the first profile
at solar noon ($t_n$), a second at sunset the same day ($t_{ss}$), a third profile at sunrise the next day
($t_{sr+1}$), and a last one at solar noon the next day ($t_{n+1}$). The sampling cycle is repeated every
four days. The fIon cycle is performed over a single 24-hour period; it begins at sunrise ($t_{sr}$),
followed by a second profile at solar noon ($t_n$), a third at sunset ($t_{ss}$) and a last night profile at
approximately midnight ($t_m$). For this float, the sampling cycle is repeated each day.
**2.3     Characterization of the diel cycle of the bio-optical properties**

In order to characterize the amplitude and variability of the diel cycle of the $c_p$ and $b_{bp}$

coefficients, we use the metrics defined by Kheireddine & Antoine (2014). We compute the
relative daily variation $\tilde{\Delta}c_p$ and $\tilde{\Delta}b_{bp}$ (expressed as % change) for each float and each day of
observation, from sunrise to sunrise (fIon) as follows:
$$\tilde{\Delta}c_p = 100 \left( \frac{c_p(t_{sr})}{c_p(t_{sr+1})} - 1 \right) \qquad\qquad (3)$$
$$\tilde{\Delta}b_{bp} = 100 \left( \frac{b_{bp}(t_{sr})}{b_{bp}(t_{sr+1})} - 1 \right) \qquad\qquad (4)$$
with $c_p(t_{sr})$ and $b_{bp}(t_{sr})$ being the values of $c_p$ and $b_{bp}$ at sunrise and $c_p(t_{sr+1})$ and
$b_{bp}(t_{sr+1})$ the values at sunrise the next day. Comparable computations for fLig were made





from noon ($t_n$) to noon ($t_{n+1}$). Then the mean and range in relative daily variations ($\widetilde{m\Delta}$ and $\widetilde{r\Delta}$,
respectively) are computed for each float over the entire time series.

### 2.4 Principle of the bio-optical diel cycle-based approach to biological production

In this study we considered two distinct bio-optical properties, i.e. the $c_p$ coefficient
(Siegel et al. 1989; Claustre et al. 2008) and the $b_{bp}$ coefficient (Barnes & Antoine 2014;
Kheireddine & Antoine 2014). To first order, $c_p$ and $b_{bp}$ are linearly correlated to, and thus
may be used as a proxy for, the stock of POC (e.g. Oubelkheir et al. 2005; Gardner et al.
2006; Cetinić et al. 2012). Both of these bio-optical proxies have been shown to exhibit a
diurnal cycle (e.g. Oubelkheir & Sciandra 2008; Loisel e al. 2011; Kheireddine & Antoine

2014).

The daily solar cycle is a major driver of biological activity in all oceanic euphotic
zones (e.g. Oubelkheir & Sciandra 2008), which influences the abundance of microorganisms
(Jacquet et al. 1998; Vaulot & Marie 1999; Brunet et al. 2007) and therefore, the magnitude
of the $c_p$ and $b_{bp}$ coefficients. To first order, the diurnal increase in $c_p$ or $b_{bp}$ may be attributed
to an increase in particle biomass, resulting from an increase in particle abundance and/or
size, both associated with cell division. To second order, the diurnal increase in $c_p$ or $b_{bp}$ may
be caused by variations in particle shape and refractive index (Stramski & Reynolds 1993;
Durand & Olson 1996; Claustre et al. 2002; Durand et al. 2002). The night decrease in $c_p$ or
$b_{bp}$ may be explained by a decrease in particle abundance due to particle aggregation and
sinking or grazing (Cullen et al. 1992), changes in particle size and/or refractive index
associated with cell division, or microorganism respiration.
The bio-optical diel cycle-based approach used in this study relies on a modified version
of Claustre et al. (2008). Following this approach, the observed daytime increase and
nighttime decrease in $c_p$-derived (or $b_{bp}$-derived) POC are used to estimate gross community





production. For this purpose, the $c_p$ and $b_{bp}$ coefficients, measured *in situ* by the BGC-Argo
profiling floats, are converted into POC equivalent using a constant $c_p$-to-POC (or $b_{bp}$-to-
POC) relationship from the literature (see below). By definition, the $c_p$ and $b_{bp}$ coefficients
target particles so that the dissolved biological matter is not accounted for by the present
method.

### 2.5    Bio-optical properties-to-POC relationships

The conversion of $c_p$ and $b_{bp}$ into POC relies on the use of empirical proxy relationships
and assumptions concerning the variations in those relationships. First, as in Claustre et al.
(2008), we assume that the $c_p$- or $b_{bp}$-to-POC relationship remains constant on a daily
timescale, consistently with previous works (Stramski & Reynolds 1993; Cullen & Lewis
1995), so that observed variations in the optical coefficients can be interpreted as variations in
POC. Second, the specific proxy value is not constant, as many empirical relationships
between POC and $c_p$ (e.g. Claustre et al. 1999; Oubelkheir et al. 2005; Gardner et al. 2006;
Loisel et al. 2011) or $b_{bp}$ (e.g. Stramski et al. 2008; Loisel et al. 2011; Cetinić et al. 2012)
have been proposed for specific regions (Tables 1 and 2). In the present study, we used the
relationships from Oubelkheir et al. (2005) and Loisel et al. (2011) for $c_p$ and $b_{bp}$,
respectively. Both relationships were established from *in situ* measurements collected in the
Mediterranean Sea and produce $c_p$- or $b_{bp}$-derived POC values falling in the middle of the
range of all the POC values resulting from the different bio-optical relationships taken from
the literature (Tables 1 and 2).

### 2.6    Estimating biological production from the diel cycle of POC

#### 2.6.1    Hypotheses

The time-rate-of-change in depth-resolved POC biomass, b(z,t), can be described by a
partial differential equation:



$\frac{\partial b(z,t)}{\partial t} = \mu(z,t)\,b(z,t) - l(z,t)\,b(z,t),$  (5)
where $\mu(z,t)$ is the particle photosynthetic growth rate and $l(z,t)$ the particle loss rate at
depth z and time t (both in units of $d^{-1}$). As in previous studies (Claustre et al. 2008, Gernez et
al. 2011; Barnes and Antoine 2014), we assume a quasi-1D framework. In other words, we
ignore the effects of lateral transport of particles by oceanic currents and assume that there is
no vertical transport of particles in and out the layer considered. We also assume that the loss
rate is constant throughout the day and uniform with depth, i.e. $l(z,t) = l$. In this context, the
time series of profiles are first converted to depth-integrated biomass (from b(z,t) to B(t)) for
each of the layers in question, and then integrated over time to determine daytime gain,
nighttime loss, and net daily production.
**2.6.2  Calculation of the loss rate**
During nighttime, there is no photosynthetic growth, so that Eq. (5) becomes:
$\frac{\partial b(z,t)}{\partial t} = l\,b(z,t).$  (6)
The integration of Eq. (6) over depth yields an expression of the rate of change of the depth-
integrated POC biomass, B(t):
$\frac{\partial B(t)}{\partial t} = -\,l\,B(t),$  (7)
with $B(t) = \int_{z_2}^{z_1} b(z,t)dz$, the POC integrated within a given layer of the water column,
comprised between the depths $z_1$ and $z_2$ (in gC m$^{-2}$). In this respect, we consider three
different layers: the euphotic layer extending from $z_1 = 0$ m to $z_2 = Z_{eu}$; the surface layer
extending from $z_1 = 0$ m to $z_2 = Z_{pd}$; and the SCM layer extending from $z_1 = Z_{SCM} - Z_{SCM,1/2}$
and $z_2 = Z_{SCM} + Z_{SCM,1/2}$, with $Z_{SCM}$ the depth of the SCM and $Z_{SCM,1/2}$ the depth at which Chl
is half of the SCM value.



Eq. (7) can be integrated over nighttime to obtain an equation for the loss rate $l$, as a
function of the nocturnal variation of B:
$$l = \frac{\ln \left(\frac{B_{ss}}{B_{sr+1}}\right)}{t_{sr+1} - t_{ss}},$$    (8)
with $B(t_{ss})$ and $B(t_{sr+1})$ corresponding to the POC integrated within the layer of interest, at
$t_{ss}$ (sunset) and $t_{sr+1}$ (sunrise of the next day).

### 2.6.3   Calculation of the production rate

The daily and depth-integrated production of particles, P (in units of gC m$^{-2}$ d$^{-1}$), is
defined as:
$$P = \int_{t_{sr}}^{t_{sr+1}} \int_{z2}^{z1} \mu(z,t) \, b(z,t) \, dz \, dt,$$    (9)
with $t_{sr}$ the time of sunrise on day 1 and $t_{sr+1}$ the time of sunrise the following day. Equation
(5) can be used to express P as a function of $l$, b(z,t), and the rate of change of b(z,t):
$$P = \int_{t_{sr}}^{t_{sr+1}} \int_{z2}^{z1} \left(\frac{\partial b(z,t)}{\partial t} + l \, b(z,t)\right) dz \, dt,$$    (10)
which yields:
$$P = B_{t_{sr+1}} - B_{t_{sr}} + l \int_{t_{sr}}^{t_{sr+1}} B(t) \, dt.$$    (11)
Finally, using the trapezoidal rule, Eq. (11) simplifies into
$$P = B_{t_{sr+1}} - B_{t_{sr}} + l \sum_{i=1}^{j} (t_{i+1} - t_i) \frac{B_{i+1} + B_i}{2},$$    (12)
with $l$ calculated from Eq. (8) and the index i corresponding to the different measurement time
steps over the course of the diel cycle ($t_n$, $t_{ss}$, $t_{sr+1}$, and $t_{n+1}$ for the Ligurian Sea, and $t_{sr}$, $t_n$, $t_{ss}$,
and $t_m$ for the Ionian Sea; Fig. 2).
In brief, Eq. (12) is applied to the time series of the BGC-Argo floats by using $b_{bp}$ and
$c_p$ converted into POC equivalents, integrated within the euphotic, surface, and SCM layers to



compute $c_p$- and $b_{bp}$-derived estimates of gross community production, P, in all three layers of
the water column.

## 2.7  Primary production model

The community production estimates obtained from the bio-optical diel cycle-based
method are evaluated against primary production values computed with the bio-optical
primary production model of Morel (1991). Morel's model estimates the daily depth-resolved
organic carbon concentration fixed by photosynthesis, using the noontime measurements of
Chl, temperature and PAR within the water column by the BGC-Argo profiling floats as
model inputs. The standard phytoplankton photophysiological parameterization is used for
these calculations (Morel 1991; Morel et al. 1996).

## 2.8  Phytoplankton pigments and community composition

During the BOUSSOLE cruises conducted in 2014 (cruises #143 to #154) and the
PEACETIME cruise, discrete seawater samples were taken at 10–12 depths within the water
column from Niskin bottles mounted on a CTD-rosette system and then filtered under low
vacuum onto Whatman GF/F filters (0.7-µm nominal pore size, 25-mm diameter). The filters
were flash-frozen in liquid nitrogen and stored at -80°C until analysis by HPLC following the
protocol of Ras et al. (2008). The concentrations of phytoplankton pigments resulting from
these analyses were used to estimate the composition of the phytoplankton assemblage. For
this purpose, we used the diagnostic pigment-based approach (Claustre et al. 1994; Vidussi et
al. 2001; Uitz et al. 2006) with the coefficients of Di Cicco et al. (2017) to account for the
specificities of Mediterranean phytoplankton communities. This approach yields the relative
contribution to chlorophyll *a* biomass of major taxonomic groups merged into three size
classes (micro-, nano and picophytoplankton).





The fLig float was spatially distanced from the location of sampling at the BOUSSOLE
mooring site. Thus it was necessary to identify the time shift for matching the cruise-sampled
analyses to the float profile measurements. This was achieved by performing a cross-
correlation analysis of the bio-optical time series measurements collected on the float with
that on the mooring (in this case Chl, $c_p$ and $b_{bp}$). A positive time lag between the
BOUSSOLE site and the position of the fLig float during its drift is observed suggesting that
the variations observed by the float led that observed at BOUSSOLE by ~2 days. This small-
time lag coupled with high correlation coefficient values and long decorrelation time scales,
indicate that the monthly interpolated pigment data measured at the BOUSSOLE site may be
considered as representative of the pigment composition along the fLig float trajectory.
**3    Results and discussion**
In this section, we first provide an overview of the biogeochemical and bio-optical
characteristics measured by the two BGC-Argo profiling floats in the Ligurian and Ionian
Seas. We then assess the usefulness of the diel cycle of the $b_{bp}$ coefficient for deriving
community production, in comparison to the $c_p$-derived estimates as a reference, and discuss
the $c_p$-derived estimates. Finally, we examine the community production estimates in both
study regions, with an emphasis on the SCM layer and its biogeochemical significance.
**3.1    Biogeochemical and bio-optical context in the study regions**
Both study regions are characterized by either seasonal or persistent oligotrophy, with
mean surface Chl values ranging within 0.08–0.22 mg m$^{-3}$ (Fig. 3), and a stratified water
column with a consistently shallow MLD (<30 m). They do exhibit very different euphotic
zones, with a mean $Z_{eu}$ of 47±5 m and 89±4 m in the Ligurian and Ionian Seas, respectively.
Both regions also display a SCM, the depth of which co-occurs with $Z_{eu}$ and the isopycnal
28.85 (i.e. the isoline of potential density 28.85 kg m$^{-3}$) over the considered time series,



except for the last month of observation in the Ionian Sea. In the Ligurian Sea, the SCM is
intense ($1.06\pm0.34$ mg Chl m$^{-3}$; Fig. 3a), relatively shallow ($41\pm7$ m), and associated with the
subsurface $c_p$ and $b_{bp}$ maxima ($0.27\pm0.09$ and $0.0015\pm0.0006$ m$^{-1}$, respectively; Fig. 3b–c).
The Chl, and $c_p$ values are 5 times larger in the SCM layer than at surface, and the $b_{bp}$ values
3.6 times larger. In contrast, in the Ionian Sea, the SCM is associated with lower values of Chl
($0.27\pm0.07$ mg m$^{-3}$; Fig. 3d), $c_p$ ($0.05\pm0.01$ m$^{-1}$; Fig. 3e) and $b_{bp}$ ($0.0005\pm0.0001$ m$^{-1}$; Fig. 3f).
Compared to the Ligurian Sea SCM, the Ionian Sea SCM is located twice as deep ($97\pm11$ m)
and is uncoupled from any $c_p$ and $b_{bp}$ maxima that occur at shallower depth. Hence, the
selected regions are representative of two contrasted SCM systems with distinct degree of
oligotrophy, consistent with our expectations (e.g. D'Ortenzio & Ribera D'Alcalà 2009;
Barbieux et al. 2019). These observations indeed suggest that the Ligurian Sea SCM mirrors
an increase in phytoplankton carbon biomass (SBM) likely resulting from favorable light and
nutrient conditions, whereas the Ionian SCM is induced by photoacclimation of
phytoplankton cells.
Although the summer period is typically considered stable, some temporal variations
are observed over the time series that are more pronounced in the SCM layer than at surface.
In the Ligurian Sea SCM, the Chl, $c_p$ and $b_{bp}$ exhibit similar temporal evolution, with
relatively high values in late May 2014, followed by a marked decrease until mid-July (Figs.
4a–c). Then we observe two minima in Chl, $c_p$ and $b_{bp}$ that delineate a second peak between
July 14 and August 16, 2014 (as indicated by the dashed lines in Fig. 4a–c). In the Ionian Sea
SCM, the Chl, $c_p$ and $b_{bp}$ values all decrease from late May until a minimum is reached on
August 11, 2017 (dashed line in Figs. 4d–e) and a second increase is recorded later in the
season. These temporal patterns are further discussed (Section 3.4).



### 3.2 Assessment of the method

#### 3.2.1 Analysis of the diel cycle of the $b_{bp}$ coefficient

Diel cycles, characterized by a daytime increase and a nighttime decrease, are observed in both $c_p$ and $b_{bp}$ time series in all three layers of the water column (Fig. 5). In the surface layer of the Ligurian Sea, the diel cycles of $c_p$ and $b_{bp}$ exhibit, respectively, mean relative daily variation ($\widetilde{m\Delta}$) of 12.7% and 2.3%, and a range in relative daily variations ($\widetilde{r\Delta}$) of 256.7% and 28.5% (Table 3). These values are of the same order of magnitude as those reported by Kheireddine & Antoine (2014), acquired from the BOUSSOLE surface mooring in the same area and during the oligotrophic season (from -5% to 25% for $c_p$ and from -2% to 10% for $b_{bp}$). Interestingly, the diel cycle of the $c_p$ coefficient appears systematically more pronounced than that of $b_{bp}$, with larger values of $\widetilde{m\Delta}$ and $\widetilde{r\Delta}$, regardless of the considered region and layer of the water column (Table 3).

To first order, the variability in the $b_{bp}$ and $c_p$ coefficients is determined by the variability in particle concentration, which underpins their robustness as POC proxies in open-ocean conditions and explains their coherent evolution on a monthly timescale (Figs. 3–4). Nevertheless, to second order, these coefficients vary differentially with the size and composition of the particle pool. In particular, phytoplankton make a larger contribution to $c_p$ than $b_{bp}$, in part due to their strong absorption efficient. In addition, $b_{bp}$ is more sensitive to smaller (<1 µm) particles (Stramski & Kiefer 1991; Ahn et al. 1992; Stramski et al. 2001; Boss et al. 2004) and to particle shape and internal structure (Bernard et al. 2009; Neukermans et al. 2012; Moutier et al. 2017; Organelli et al. 2018) than is $c_p$. While the diel cycle of $c_p$ would be essentially driven by photosynthetic processes due to the influence of phytoplankton on $c_p$, $b_{bp}$ would be more responsive to detritus and/or heterotrophic bacteria that show minor, if not negligible, daily variability. Hence, such specificities in the bio-optical coefficients may explain the observed differences in their diel cycles.





Based on high-frequency surface measurements in the Ligurian Sea in various seasons,
the studies of Kheireddine & Antoine (2014) and Barnes & Antoine (2014) demonstrated that
the diel cycle of $b_{bp}$ not only exhibits much reduced relative amplitude compared to that of $c_P$,
but the features of the $b_{bp}$ cycle are not synchronous with that of the $c_p$ cycle. Thus $b_{bp}$ cannot
be used interchangeably with $c_p$ for assessing daily changes in POC or community production.
Our results support these previous findings, not only for the surface layer of the Ligurian Sea,
but also for the whole water column of both the Ligurian and Ionian regions.
We now consider the integrated euphotic zone gross community production estimates
derived from the bio-optical diel cycle-based method (Fig. 6). We compare the $c_p$- and $b_{bp}$-
based estimates with primary production estimates computed with the model of Morel (1991).
The $b_{bp}$-derived production rates underestimate those derived from $c_p$ in both regions by about
a factor of ten, with respective mean values of $0.11\pm0.28$ gC m$^{-2}$ d$^{-1}$ and $1.18 \pm1.13$ gC m$^{-2}$ d$^{-1}$
in the Ligurian Sea, and $0.04\pm0.04$ gC m$^{-2}$ d$^{-1}$ and $0.46 \pm0.11$ gC m$^{-2}$ d$^{-1}$ in the Ionian Sea. In
addition, the $b_{bp}$-derived production is much lower than the primary production computed
with the model of Morel (1991), which has mean values of $0.91\pm0.14$ gC m$^{-2}$ d$^{-1}$ in the
Ligurian Sea and $0.31 \pm0.04$ gC m$^{-2}$ d$^{-1}$ in the Ionian Sea. The significantly lower community
production rates are a direct effect of the dampened relative daily amplitude of the $b_{bp}$ diel
cycle (Table 3), and the sensitivity of $b_{bp}$ to the smaller heterotrophic and detrital particulate
matter. These bio-optical diel cycle-based method, whether applied to $c_p$ or $b_{bp}$, yields an
estimate of the community production, i.e. that associated with the accumulation of
phytoplankton *and* bacteria biomass, which is necessarily larger than the primary (photo-
autotrophic) production rates from the Morel (1991) model. These questionable low values of
community production, along with the observation of a weak daily variability in $b_{bp}$, support
the idea that the diel cycle of $b_{bp}$ may not be a reliable index for total community production
rates, consistently with previous studies (Kheireddine & Antoine 2014; Barnes & Antoine





2014). However, the utility of a $b_{bp}$-derived community production may be revealed in
elucidating rates for distinct size-based groups of organisms, such as picoplankton. A better
understanding of the specific size range that dominates the diel cycle in $b_{bp}$ will be important
to understand. Yet, for our purposes, we disregard the $b_{bp}$-based estimates and focus our
analysis on the $c_p$-derived gross community production estimates.

### 3.2.2   Community production derived from the $c_p$ coefficient

The $c_p$-derived estimates of gross community production, integrated within the euphotic

layer, compare favorably with those found in the literature for similar Mediterranean areas
(see Table 4 and references therein). The $c_p$-based estimates show a 2.5-fold difference
between the Ligurian Sea and the Ionian Sea (mean of 1.18 gC m$^{-2}$ d$^{-1}$ and 0.46 gC m$^{-2}$ d$^{-1}$,
respectively; Table 6). In comparison, water column-integrated primary production values,
either inferred from satellite observations and biogeochemical models or measured *in situ*,
vary within the range 0.13–1 gC m$^{-2}$ d$^{-1}$ and 0.14–0.69 gC m$^{-2}$ d$^{-1}$ for the Western (or
Ligurian) and Eastern (or Ionian) region, respectively (Table 4). As expected, our $c_p$-based
community production rates are larger than published primary production rates. The present
$c_p$-derived values also compare favorably with gross community production estimates inferred
from a similar approach applied to bio-optical measurements from the BOUSSOLE mooring
in the Ligurian Sea (0.8–1.5 gC m$^{-2}$ d$^{-1}$ in May–August; Barnes & Antoine 2014) and along an
oligotrophic gradient in the South Pacific Subtropical Ocean (0.85 gC m$^{-2}$ d$^{-1}$; Claustre et al.

2008).

The empirical relationships linking the $c_p$ (or $b_{bp}$) coefficient to POC are known to exhibit

regional and seasonal variability in response to changes in the composition of the particle
assemblage and associated changes in particle size, shape and type, i.e. biogenic or mineral
(e.g. Stramski et al. 2004; Neukermans et al. 2012; Slade & Boss 2015). Hence, the choice of
such relationships strongly affects the conversion of the measured daily bio-optical variability



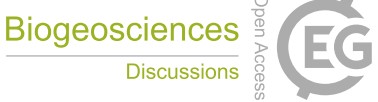

into POC fluxes. For the time period and study regions here, the $c_p$-based community
production varies by a factor of 2, depending on the selected bio-optical relationship, so that
$c_p$-based estimates vary between $0.89\pm0.84$ gC m$^{-2}$ d$^{-1}$ and $1.62\pm1.54$ gC m$^{-2}$ d$^{-1}$ in the
Ligurian Sea, and between $0.35\pm0.09$ gC m$^{-2}$ d$^{-1}$ and $0.63\pm0.16$ gC m$^{-2}$ d$^{-1}$ in the Ionian Sea.
The minimal and maximal values are obtained with the bio-optical relationships from Marra
et al. (1995) and Stramski et al. (2008), respectively (Table 5). Compared to the reference
value obtained using the Oubelkheir et al. (2005) relationship, the $c_p$-based estimates are 25%
lower and 37% higher using the relationships of Marra et al. (1995) and Stramski et al.
(2008), respectively.

The use of the single relationship established from Mediterranean waters (Oubelkheir et

al. 2005) appears as a reasonable choice for the study region. Yet, if more relevant bio-optical
proxy relationships are available, such as one that accounts for spatial and seasonal variations,
and even applicable to different layers of the water column, that would certainly reduce the
uncertainty in the rate estimation. Although this is beyond the scope of the present study, we
recognize that such investigations should be conducted in the future in order to refine optics-
based biomass (POC) and community production estimates.

### 3.3    Regional and vertical variability of production

The temporal evolution of the $c_p$-derived POC biomass integrated within the three

distinct layers of the water column is presented for the two study regions in Fig. 7. The
integrated POC concentration values follow similar temporal trends as reported for $c_p$ (Figs.
3–4). In the Ligurian Sea, the euphotic layer-integrated POC varies between 1.5 and 6.0 gC
m$^{-2}$ (mean of $3.7\pm1.1$ gC m$^{-2}$; Fig. 7a and Table 6). There was a decrease from late May to
mid-July (6.0 to 1.5 gC m$^{-2}$) followed by a moderate peak (3.9 gC m$^{-2}$) between mid-July and
mid-August (as bounded by the dashed lines in Fig. 5). The $c_p$-based community production
did exhibit large variability over the time period (Fig. 7b and Table 6), but interestingly, the





moderate POC peak observed in the core of the oligotrophic season (between mid-July and
mid-August) is associated with the maximum production rate of the time series (4.3 gC m$^{-2}$ d$^{-1}$).

In the Ionian Sea, the POC biomass integrated within the euphotic zone is much lower

than in the Ligurian Sea and remains more stable over the time period (1.9±0.24 gC m$^{-2}$; Fig.
7c and Table 6). As with POC, the community production is much lower in the Ionian Sea
than in the Ligurian Sea, but still exhibits substantial variability with values ranging within
0.06–0.68 gC m$^{-2}$ d$^{-1}$ (Fig. 7d). These results are consistent with multiple studies reporting a
large difference in the trophic status and productivity of the Ligurian and Ionian Seas, on
seasonal and annual timescales (D'Ortenzio & Ribera d'Alcala, 2009; Siokou-Frangou et al.
2010; Lavigne et al. 2013; Mayot et al. 2016). Our results confirm this difference, yet on a
monthly timescale during the oligotrophic summer period.

The gross community production estimates integrated over different layers of the

water column reveal distinct patterns. In the Ligurian Sea, both the euphotic and SCM layers
show large production rates (0.96±1.3 gC m$^{-2}$ d$^{-1}$), with production in the SCM layer
frequently equaling or overtaking on the production in the euphotic layer (Fig. 7b). This is
particularly striking in late July, when the production peak is actually associated with a large
enhancement of the production in the SCM layer (4.9 gC m$^{-2}$ d$^{-1}$). In contrast, the surface
layer shows reduced production rates (0.29±0.33 gC m$^{-2}$ d$^{-1}$), a pattern also observed in the
Ionian Sea (0.11±0.04 gC m$^{-2}$ d$^{-1}$). In the Ionian Sea, the production is maximal in the
euphotic zone, and very variable and occasionally larger in the SCM layer (0.14±0.39 gC m$^{-2}$
d$^{-1}$; Fig. 7d). The bio-optical diel cycle-based method produces several occurrences of
negative values in the SCM layer, indicating that the quasi-1-D assumption is occasionally not
satisfied in the lower part of the euphotic layer. This could arise when physical processes that
transport particles are larger than local growth and loss of POC.



Our results support the hypothesis raised in previous studies (e.g. Mignot et al. 2014;
Barbieux et al. 2019) that, in the Ligurian temperate-like system, the SCM, which is in fact an
SBM, may be highly productive. Conversely, in the Ionian region, which shows similarities
with subtropical stratified oligotrophic systems, the SCM reflects photoacclimation and is less
productive. Beyond these mean regional trends, both SCM systems exhibit some temporal
variability in production, a somewhat unexpected pattern at the core of the presumably stable
oligotrophic season.
**3.4    Production in the SCM layer in relation with the biotic and abiotic context**
Here we investigate the temporal variability in the SCM layer production and attempt to
interpret the observed patterns in the context of biological and abiotic conditions.
**3.4.1    Phytoplankton and particulate assemblage**
The pigment data collected during the BOUSSOLE and PEACETIME cruises
concomitantly with the deployments of the fLig and fIon floats, respectively, are used as
proxies for phytoplankton community structure (Fig. 8). In the Ligurian Sea,
nanophytoplankton (mainly prymnesiophytes) appear as dominant contributors to the
phytoplankton assemblage both in the surface layer (48±8%; Fig. 8b) and SCM layer
(54±10%). Picophytoplankton (prokaryotes and small chlorophytes) and microphytoplankton
(diatoms and dinoflagellates) are present in moderate proportions, with 30±11% and 22±5%
in the upper layer, and 19±7% and 27±9% in the SCM layer, respectively (Figs. 8a and 8c).
No marked community change is observed during the timeseries. In the Ionian Sea, the
surface layer displays large contribution of nanophytoplankton (56±2%; Fig. 8e) and, to a
lesser extent, picophytoplankton (29±3%; Fig. 8d). However, the SCM level is characterized
by an enhanced contribution of microphytoplankton (diatoms) to the algal assemblage
(49±5%; Fig. 8f), as discussed in Marañon et al. (2021). The Ionian PEACETIME data was





limited to the period from May 25 to 28, 2017, and thus it was not possible to determine
whether the composition of phytoplankton communities evolved with time. Although not
characterized by the prokaryotic populations (*Synechococcus* and *Prochlorococcus*) that
typically prevail in stratified oligotrophic environments, our observations are consistent with
previous studies reporting enhanced contributions of nanophytoplankton (e.g. Gitelson et al.
1996; Vidussi et al. 2001) and the occurrence of diatoms at depth (Siokou-Frangou et al.
2010; Crombet et al. 2011; Marañon et al. 2021) in the Mediterranean Sea.
Bio-optical properties and their ratios provide indication about variations in the
constituents (algal or nonalgal) and size of the particulate pool, the composition of the
phytoplankton assemblage and the physiological status of phytoplankton cells (e.g. Geider
1987; Ulloa et al. 1994; Stramski et al. 2004; Loisel et al. 2007). Here we consider the bio-
optical ratios $b_{bp} / c_p$, $c_p / Chl$, and $b_{bp} / Chl$ in the SCM layer (Fig. 9). The $b_{bp} / c_p$ ratio, while
at slightly different wavelengths (700 nm and 660 nm, respectively) are at absorption minima
and thus this ratio is comparable to the backscattering ratio $b_{bp} / b_p$. The $b_{bp} / b_p$ ratio is a
demonstrated proxy for determining relative constituent composition (Twardowski et al.
2001), with phytoplankton exhibiting lower ratios than nonalgal particles (approximately
0.5% and 1%, respectively; Boss et al. 2004; Whitmire et al. 2007; Westberry et al. 2010).
The $b_{bp} / Chl$ and $c_p / Chl$ ratios are both proxies for the POC / Chl ratio (e.g. Claustre et al.
1999; Oubelkheir et al. 2005; Behrenfeld et al. 2015; Álvarez et al. 2016), and thus an
indicator of the contribution of phytoplankton to the whole organic pool. The variations are
also interpreted as changes in the composition of phytoplankton communities (e.g.
Sathyendranath et al. 2009) and their acclimation to the light-nutrient regime (e.g. Geider et
al. 1987; Geider et al. 1997; Cloern 1999) if one assumes that nonalgal particles are negligible
(e.g., as indicated by the backscattering ratio) or not varying in concentration. The differences
between the $b_{bp} / Chl$ and $c_p / Chl$ ratios lie in the fact that they are sensitive to different





particle size ranges (Roesler and Boss 2008) and, thus, when they are not correlated, one can
qualitatively discern differing dynamics across the phytoplankton size spectrum.
The $b_{bp}$ / $c_p$ ratio is very different between the Ligurian and Ionian Seas, with significantly
lower values in the Ligurian Sea ($0.0068\pm0.0009$ or $0.68\pm0.09\%$, and $0.0095\pm0.0009$ or
$0.95\pm0.09\%$; Fig. 9). These ratios indicate that, in the general sense, the Ligurian Sea SCM is
more phytoplankton dominated than the Ionian Sea SCM, which tends towards nonalgal
particles. In the Ligurian Sea, the $b_{bp}$ / $c_p$ ratio remains $<0.0087$ ($0.87\%$) and reaches a
minimum of 0.0055 ($0.55\%$) over the period coinciding with the production event from mid-
July to mid-August (Fig. 9a), consistent with phytoplankton dominance. In contrast, in the
Ionian Sea SCM, the $b_{bp}$ / $c_p$ ratio increases from 0.0085 ($0.85\%$) in late May, peaking at
nearly 0.012 ($1.2\%$) in early August, and then decreasing back to 0.0085 ($0.85\%$) in
September (Fig. 9b). The tendency towards a ratio of $1\%$ in the core of the oligotrophic
season, evidences the increased proportion of nonalgal particles to the bulk pool as previously
observed in oligotrophic environments (Yentsch & Phinney 1989; Stramski et al. 2004; Loisel
et al. 2007).
The $c_p$ and $b_{bp}$ to Chl ratios exhibit not only different temporal patterns between the
Ligurian and Ionian Sea SCMs, they also exhibit different relative values. The $c_p$ / Chl ratio in
the Ligurian Sea SCM is higher than that of the Ionian Sea, ranging from 0.18 to 0.45 $m^2$ mg
$Chl^{-1}$, compared to 0.15 to 0.26 $m^2$ mg $Chl^{-1}$, respectively. In contrast, although the $b_{bp}$ / Chl
ratio in the Ligurian Sea SCM ranges from 0.0011 to 0.0023 $m^2$ mg $Chl^{-1}$, and the Ionian Sea
from 0.0015 to 0.0021 $m^2$ mg $Chl^{-1}$, they have essentially identical mean values over the time
series ($0.0017\pm0.0006$ and $0.0017\pm0.0001$, respectively). This suggests that the POC in the
smaller size fractions are more similar in their respective SCM than in their larger size
fractions.





Temporally, the Ligurian Sea SCM exhibits significantly more temporal variations in both
ratios compared to the Ionian Sea SCM, and the temporal variations are highly correlated.
Both the $c_p$ / Chl and $b_{bp}$ / Chl ratios in the Ligurian Sea SCM exhibit a peak at the start of the
time series in late May that decreases to mid-July, followed by a second peak during the
period coinciding with the production episode from mid-July to mid-August, and then a third
increase until the end of the time series (Figs. 9b–c). In contrast, both ratios in the Ionian Sea
SCM exhibit significantly reduced temporal variability (Figs. 9e–f), with a weak increase is
observed starting in early August.
Despite differing temporal variability, the $b_{bp}$ / Chl ratio in both Seas remains moderate to
low (<0.0025 $m^2$ mg Chl$^{-1}$; Figs. 9c and 9f, consistent with global SCM values (Barbieux et
al., 2018). The enhanced $b_{bp}$ / Chl values observed in the Ligurian Sea SCM in early May, late
July and late August suggest an increased contribution of small (pico- and nano-sized)
phytoplankton (Cetinić et al. 2012; Cetinić et al. 2015). Yet, the BOUSSOLE pigment data do
not reveal pronounced changes in the phytoplankton assemblage. Low-light conditions
typically prevailing in the SCM layer are usually associated with low values of the $c_p$ / Chl
and $b_{bp}$ / Chl ratios (e.g. (Behrenfeld & Boss 2003; Westberry et al., 2008; Barbieux et al.
2019), caused by photoacclimation by which phytoplankton organisms increase their
intracellular Chl. Nevertheless, the temporal variability in the $c_p$ / Chl and $b_{bp}$ / Chl values
may be resulting from fluctuations in the light conditions at the SCM in the Ligurian Sea. In
the Ionian Sea, the invariant low $c_p$ / Chl and $b_{bp}$ / Chl values are consistent with both
photoacclimation of phytoplankton to low-light conditions and a diatom-dominated
phytoplankton assemblage (Cetinić et al. 2015; Barbieux et al. 2018). The relatively stable
ratios observed in this region suggest a relative steadiness in the composition of the
phytoplankton assemblage over the considered period.



### 3.4.2 Relation to abiotic conditions

The Ligurian Sea exhibits enhanced community production during the period from mid-July to mid-August 2014, which is associated with a comparatively moderate increase in the biomass indicators (Figs. 3–4) and $c_p$-derived POC (Fig. 7a). During this time period, the depth of the SCM shoals by 25 m. This change occurs concurrently with a slight shoaling of the density isopycnals (Figs. 3a–c), and a doubling (from 0.5 to 1 mol quanta m$^{-2}$ d$^{-1}$) in the daily PAR within the SCM layer (Fig. 10a). Therefore, we suggest that the observed production episode may result from physical forcing that induces an upwelling of the water mass, thereby resulting in an alleviation of the light/nutrient limitation and an adequate balance between light and nutrient availability in the SCM layer. This SCM production episode is associated with a moderate phytoplankton biomass (0.8 Chl mg m$^{-3}$), dominated by a nanoplankton community. It coincides with an increase in the $c_p$ / Chl and $b_{bp}$ / Chl ratios, which we attribute to a boost in the carbon-to-Chl ratio resulting from production in enhanced light conditions. Because it appears to result from changes in light conditions, we may attribute this production event to phytoplankton (not community) growth.

In the Ionian Sea, the depth of the SCM follows the depth of the isopycnal 28.85 during the period from late to May to mid-August 2017 (Figs. 3d–f). In mid-August, the SCM reaches its deepest point (~125 m) concurrent with deepening isopycnals, decreased PAR levels within the SCM layer (Fig. 10b) and minimum values of Chl, $c_p$ and $b_{bp}$. Afterwards, the SCM depth decouples from the position of the isopycnals (Fig. 3d–f), the SCM becomes shallower and the mean daily PAR in the SCM layer increases. Nevertheless, the observed temporal fluctuations in the abiotic forcing and biological indicators do not seem to relate with any clear change in the community production (Figs. 7d–f). This suggests that physics-induced changes in the position of the SCM are not sufficient to alleviate the light and/or nutrient limitation occurring at this time in the considered area (Guieu et al. 2020).





Considering the large contribution of diatoms at the SCM, one may conclude that the low, yet
non-negligible, production levels estimated in the SCM layer are supported by diatoms. This
result supports previous findings indicating that, contrary to the classic view of diatoms
thriving essentially in dynamic eutrophic conditions, these organisms have the ability to
maintain in stratified oligotrophic environments, including in deep layers under low light-
nutrient conditions (Kemp & Villareal 2013; Kemp & Villareal, 2018).
**3.5    Contribution of the SCM to the water column production**

In order to assess the relative contribution of the SCM layer to the production occurring

in the whole water column, we compare the $c_P$-based estimates integrated within the
productive layer (0–1.5 $Z_{eu}$) and SCM layers. Our results suggest that, for these oligotrophic
systems, the production integrated within the SCM layer represents a substantial fraction
($F_{SCM}$) of the gross community production integrated within the productive layer. This is
particularly the case for the Ligurian Sea where $F_{SCM}$ reaches ~42%, and to a lesser extent for
the Ionian Sea with $F_{SCM}$ ~16%.

Subtropical stratified oligotrophic gyres cover 45% of the global ocean (McClain et al.

2004). Assuming that the Ionian Sea is representative of such systems (e.g. Mignot et al.
2014; Barbieux et al. 2019), and extrapolating the estimated relative contribution of the SCM
layer to the water column production in the Ionian ($F_{SCM}$ ~16%), then the SCM layer would
contribute ~7% of the community production of the water column on a global scale (i.e. $F_{SCM}$
of 16% multiplied by a global spatial occurrence of 45%). In addition, using a global BGC-
Argo database, Cornec et al. (2021) estimated that SCMs in oligotrophic subtropical gyres
behave as SBM 8–42% of the year, depending on the season. Thus, assuming the Ligurian
SCM oligotrophic summer system as a reference for SBM, the contribution of the SCM layer
to the global water column production could seasonally reach 19% (i.e. $F_{SCM}$ of 42%
multiplied by a global spatial occurrence of 45%).



We recognize that these estimates are very crude and need to be refined and confirmed
in future studies. Yet they suggest that the contribution of the SCM layer to the water column
production may be significant globally, although commonly ignored. Our observations are
consistent with previous findings (Crombet et al. 2011; Kemp & Villareal 2013; Mignot et al.
2014), and suggest that stratified oligotrophic systems should no longer be considered as
steady oceanic deserts and that their biogeochemical contribution should be accounted for and
better quantified to improve global carbon budgets.
**4    Conclusions**
The present study represents a first attempt to apply the bio-optical diel cycle-based
method (Siegel et al. 1989; Claustre et al. 2008) to the $c_p$ and $b_{bp}$ coefficients measured by
two BGC-Argo profiling floats. It aims to quantify gross community production in different
layers of the water column, the subsurface chlorophyll maximum (SCM) layer in particular,
during the oligotrophic summer season in two distinct systems of the Mediterranean, i.e. the
Ligurian Sea and the Ionian Sea.
From a methodological point of view, our results indicate that, compared to the $c_p$
coefficient, the diel cycle of the $b_{bp}$ coefficient is not an optimal proxy for the daily POC
variations, so that we are not confident using the $b_{bp}$-based gross community production
estimates, although it may provide a more robust proxy for the fraction of particulate matter in
the smaller size classes. Our results for the surface layer of the Ligurian Sea are consistent
with previous studies from moored observations (Kheireddine & Antoine 2014; Barnes &
Antoine 2014), and we found they are valid for the entire water column and for both the
Ligurian and Ionian Seas. These results have major implications for use of the methodology
with geostationary ocean color missions and standard BGC-Argo profiling floats that yield
only the $b_{bp}$ coefficient. The present results thus argue in favor of a frequent implementation



onto BGC-Argo floats of transmissometers ($c_p$ sensors), which provide information on a suite
of key biogeochemical variables (Claustre et al. 2020), from phytoplankton community
composition (Rembauville et al. 2017), to particle flux export (Briggs et al. 2011; Estapa et al.
2013) and, as demonstrated here, biological production (White et al. 2017; Briggs et al. 2018).

Our $c_p$-based gross community production rates compare consistently with previous

estimates from a similar approach applied to oligotrophic waters (Claustre et al. 2008; Gernez
et al. 2011; Barnes & Antoine 2014). These values are also consistent with estimates of
primary production either computed from the model of Morel (1991) coupled to the
considered BGC-Argo data, or published in the literature, although admittedly higher. This is
unsurprising because the present estimates are based on the $c_p$ coefficient, which accounts for
both autotrophic and heterotrophic organisms (not just phytoplankton). This nevertheless
raises the question of the selection of an empirical bio-optical relationship, which is key to
converting $c_p$ into POC equivalent. In the present study, the $c_p$-derived production estimates
on average decrease by 25% or increase by 37% depending on the used bio-optical
relationship, which is not negligible. Hence, we recommend POC sampling simultaneously to
BGC-Argo floats deployment. This will help to better constrain bio-optical relationships and
ultimately improve the reliability of the biomass and production estimates.

Our results indicate that both the Ligurian and Ionian Seas may sustain relatively large

levels of gross community production during the oligotrophic summer period, with a
substantial contribution by the SCM layer, a feature characteristic of oligotrophic systems that
is typically considered as steady and non-productive. In the Ligurian, the SCM behaves as a
subsurface biomass maximum (SBM) and appears to respond to episodic abiotic forcing, with
increased production rates coinciding with enhanced light availability in response to shoaling
isopycnals and SCM. In this system, the particle assemblage is dominated by phytoplankton
organisms, mainly nanoflagellates. In contrast, the Ionian SCM layer essentially reflects





photoacclimation of phytoplankton cells to prevailing environmental conditions. It does not
seem affected by modifications in abiotic forcing, although the phytoplankton assemblage
appears to be dominated by diatoms. These results agree with previous BGC-Argo-based
studies describing the occurrence and functioning of SCM systems in the global ocean
(Mignot et al. 2014; Cornec et al. 2021) and Mediterranean Sea (Lavigne et al. 2015;
Barbieux et al. 2019), and offer a first attempt to quantify biological production in such
systems. More generally, our study suggests that the contribution of the SCM layer to the
water column production varies broadly depending the considered system, whether seasonally
(~42% in the Ligurian Sea) or permanently (~16% in the Ionian Sea) oligotrophic.

Our study emphases the promising potential of BGC-Argo profiling floats for providing

a non-intrusive, high-frequency assessment of POC production within the whole water
column, which is critical in particular for applications to stratified oligotrophic environments
with recurring or permanent SCMs. The present results, based on data from two
Mediterranean environments, should be confirmed in the future through the deployment of
"multi-profiling" BGC-Argo floats in the broad, remote subtropical gyres. In such systems,
biological production is not constant but, instead, shows high temporal heterogeneity (Karl et
al. 2003; Claustre et al. 2008) that may be missed by traditional sampling, leading to a
potential underestimate of the biogeochemical impact of these systems in global carbon
budgets. Implementing such a BGC-Argo-based approach to carbon flux quantification
becomes even more important in the perspective of climate change, which is predicted to
induce an expansion of stratified oligotrophic gyres and an oligotrophication of the oceans
(Sarmiento et al. 2004) as already observed from satellite imagery (Polovina et al. 2008;
Signorini et al. 2015).



*Author contribution*    MB, JU and AB designed the work and prepared the manuscript. MB
processed the data and conducted the analyses. MB, JU and CR prepared the plots. AM and
BG developed the biological production model. AM helped with the implementation of the
model and the interpretation of the ouput data. CR contributed to the analysis of the diel bio-
optical variability, interpretation of bio-optical data and the organization of the manuscript.
HC contributed to the interpretation of the BGC-Argo data and biological production. HL
helped with the interpretation of the bio-optical data and the global extrapolation of the
results. VT and FDO contributed to the BGC-Argo float deployments and interpretation of the
physical data. AP prepared and tested the BGC-Argo floats prior to deployment and set up the
raw data stream. EL and CP developed the BGC-Argo float version used in this study and
contributed to float preparation. CS handled BGC-Argo data archiving and distribution. All
authors reviewed and approved the manuscript.

*Data availability*        The BGC-Argo profiling float data and metadata used in this paper may
be downloaded from the Argo GDAC (http://doi.org/10.17882/42182). All other original data
are available from the Argo Global Data Assembly Center (ftp://ftp.ifremer.fr/ifremer/argo).
These data were collected and made freely available by the International Argo Program and
the national programs that contribute to it (http://www.argo.ucsd.edu; https://www.ocean-
ops.org). The Argo Program is part of the Global Ocean Observing System. The
PEACETIME project pigment data are available from the SEANOE archive under the
following reference: Guieu et al., Biogeochemical dataset collected during the PEACETIME
cruise, SEANOE, https://doi.org/10.17882/75747, 2020. The BOUSSOLE program pigment
data        may        be        accessed        upon        request        (http://www.obs-
vlfr.fr/Boussole/html/boussole_data/login_form.php).



*Acknowledgement* This paper represents a contribution to the following projects:
PEACETIME (https://doi.org/10.17600/17000300), a joint initiative of the MERMEX and
ChArMEx components supported by CNRS-INSU, IFREMER, CEA, and Météo-France as
part of the program MISTRALS coordinated by INSU; PEACETIME-OC supported by the
French program CNES-TOSCA; remOcean funded by ERC (grant 246777); and NAOS
funded by ANR Equipex (grant J11R107-F). MB was funded by a PhD grant from Sorbonne
Université (Ecole Doctorale 129). Phytoplankton pigment analyses were performed at the
SAPIGH national HPLC analytical service at the Institut de la Mer de Villefranche (IMEV).
We acknowledge the captains and crew of the Téthys and Pourquoi Pas? research vessels
during the BOUSSOLE and PEACETIME cruises, as well as David Antoine, PI of the
BOUSSOLE project, and Cécile Guieu and Karine Desboeufs, PIs of the PEACETIME
project. We thank the International Argo Program and Coriolis project, which contributed to
making the data freely and publicly available. Marin Cornec is also warmly thanked for useful
discussion regarding biological production in SCM systems.

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





**Figure captions**

**Figure 1**: Trajectories of the two BGC-Argo profiling floats fLig (WMO6901776) and fIon (WMO6902828) deployed respectively in the Ligurian Sea (green) and the Ionian Sea (blue).

**Figure 2**: Schematic representation of the diel variations of the depth-integrated bio-optical properties converted to POC biomass (B) and the sampling strategies employed in the (a) Ligurian Sea and (b) Ionian Sea. The diamond-shaped symbols indicate schematically the float profile times, labeled with time stamps associated with sunrise (sr), noon (n), sunset (ss) and midnight (m), with the corresponding POC biomass estimated within the considered layer (e.g., $B(t_{sr})$, etc.). The numeric subscripts (+1, +2, +4 or +5) indicate the number of days since the first profile of the summertime time series.

**Figure 3**: Time series of Chl (a and d), $b_{bp}$ (b and e) and $c_p$ (d and f) in the Ligurian Sea (left) and the Ionian Sea (right). The Mixed Layer Depth (MLD; black line), the isopycnal 28.85, expressed as $\sigma_t$ (blue line), the euphotic depth ($Z_{eu}$; white line) and the depth of the SCM (magenta line) are superimposed onto the bio-optical timeseries. The dashed lines indicate the dates at which the $c_p$ and the $b_{bp}$ values in the SCM layer reach a minimum.

**Figure 4**: Temporal evolution of Chl (a and d), $c_{bp}$ (b and e), and $b_{bp}$ (c and f) in the surface (red) and SCM (dark green) layers for the Ligurian Sea (left) and the Ionian Sea (right). The dashed lines indicate the dates when the values of $c_p$ and $b_{bp}$ in the SCM layer reach a minimum

**Figure 5**: Example of the variations of the $c_p$ (a) and $b_{bp}$ (b) coefficients at the daily time scale in the Ionian Sea in the SCM layer during the interval from September 2 to September 6, 2017. The grey shaded area indicates the nighttime.

**Figure 6**: Comparison of the biological production integrated within the euphotic layer, derived from the diel cycle of $c_p$ (blue) or $b_{bp}$ (yellow) or computed using the bio-optical primary production model of Morel (1991) (purple) for the Ligurian Sea (a) and the Ionian Sea (b).

**Figure 7**: Temporal evolution of the POC and community production derived from the diel cycle of $c_p$ in the Ligurian Sea (a– b) and the Ionian Sea (c–d) and integrated in three different layers of the water column: surface (dark green), euphotic (light blue) and SCM (red) layers. The dotted lines indicate the dates when $c_p$ in the SCM layer reaches a minimum.



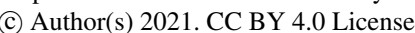


**Figure 8**: Depth-interpolated time series of the relative contributions to the chlorophyll *a* concentration (%) of the micro- (a and d), nano- (b and e) and picophytoplankton (c and h) derived from HPLC pigment determinations in the Ligurian Sea (BOUSSOLE site; left) and the Ionian Sea (PEACETIME cruise; right). The pigment data were collected at the BOUSSOLE site in the same region and at the same time period as the fLig float deployment on May 24, 2014 (see text section 2.1). The fION float was deployed concurrently to sampling for HPLC pigments at the PEACETIME ION station on May 28, 2017. Pigment data collected at ION over four days prior to float deployment are shown.

**Figure 9**: Temporal evolution of the bio-optical ratios of $b_{bp}$ / $c_p$ (a), $c_p$ / Chl (b) and $b_{bp}$ / Chl (c) in the SCM layer for the Ligurian Sea (left) and the Ionian Sea (right). The dotted lines indicate the dates when the values of $c_p$ in the SCM layer reach a minimum.

**Figure 10**: Time series of the daily-integrated photosynthetically available radiation (PAR) at the SCM level in the Ligurian Sea (a) and the Ionian Sea (b). The horizontal grey line shows the median of each time series. The dotted lines indicate the dates at which the values of $c_p$ in the SCM layer reach a minimum.



**Table 1:** POC-to-$c_p$ relationships from the literature, with POC and $c_p$ in units of mg m$^{-3}$ and m$^{-1}$, respectively.

| Reference | Region | Relationship |
|---|---|---|
| Marra et al. (1995) | North Atlantic | POC = 367 $c_p$(660) + 31.2 |
| Claustre et al. (1999) | Equatorial Pacific | POC = 501.81 $c_p$(660) + 5.33 |
| Oubelkheir et al. (2005) | Mediterranean | POC = 574 $c_p$(555) – 7.4 |
| Behrenfeld & Boss (2006) | Equatorial Pacific | POC = 585.2 $c_p$(660) + 7.6 |
| Gardner et al. (2006) | Global Ocean | POC = 381 $c_p$(660) + 9.4 |
| Stramski et al. (2008) | Pacific and Atlantic, including upwelling | POC = 661.9 $c_p$(660) – 2.168 |
| Loisel et al (2011) | Mediterranean | POC = 404 $c_{bp}$(660) + 29.25 |
| Cetinić et al. (2012) | North Atlantic | POC = 391 $c_p$(660) – 5.8 |





**Table 2:** POC-to-$b_{bp}$ relationships from the literature, with POC and $b_{bp}$ in units of mg m$^{-3}$ and m$^{-1}$, respectively.

| Reference | Region | Relationship |
|---|---|---|
| Stramski et al. (2008) | Pacific and Atlantic, including upwelling | POC = 71002 $b_{bp}$(555) –5.5 |
| Loisel et al (2011) | Mediterranean | POC = 37550 $b_{bp}$(555) + 1.3 |
| Cetinić et al. (2012) | North Atlantic | POC = 35422 $b_{bp}$(700) – 14.4 |





**Table 3:** Mean and range (%) in relative daily variations ($\widetilde{m\Delta}$ and $\widetilde{r\Delta}$, respectively) in the diel cycle of $c_p$ and $b_{bp}$ computed for each float over the entire time series, for the two considered regions and in the different layers of the water column, i.e. surface (0-$Z_{pd}$), SCM (SCM), and euphotic (0-$Z_{eu}$) layers.

| Region | | Surface layer | | SCM layer | | Euphotic layer | |
|---|---|---|---|---|---|---|---|
| | | $\tilde{\Delta}c_p$ | $\tilde{\Delta}b_{bp}$ | $\tilde{\Delta}c_p$ | $\tilde{\Delta}b_{bp}$ | $\tilde{\Delta}c_p$ | $\tilde{\Delta}b_{bp}$ |
| **Ligurian Sea** | $\widetilde{m\Delta}$ | 12.7 | -2.3 | 14.5 | 3.8 | -134.9 | 1.1 |
| | $\widetilde{r\Delta}$ | 256.7 | 28.5 | 194.8 | 107.8 | 2603.1 | 20.5 |
| **Ionian Sea** | $\widetilde{m\Delta}$ | 0.55 | 0.23 | 1.16 | 0.06 | 0.39 | 0.21 |
| | $\widetilde{r\Delta}$ | 54.4 | 21.2 | 102.4 | 57.3 | 55.9 | 24.9 |





**Table 4:** Estimates of primary and community production (in units of gC m$^{-2}$ d$^{-1}$) from the literature in areas of the Mediterranean Sea comparable, when possible, to the considered study regions.

**Primary production**

| Method | Reference | Area | Period | Layer | Estimate |
|---|---|---|---|---|---|
| Ocean color-coupled bio-optical model | Morel & André (1991) | Western basin | 1981 | 0-$Z_{eu}$ | 0.26 |
| | Antoine et al. (1995) | Whole basin | 1979-1981 | 0-1.5$Z_{eu}$ | 0.34 |
| | Bosc et al. (2004) | Western basin | 1998-2001 | 0-1.5$Z_{eu}$ | 0.45 |
| | - | Eastern basin | - | - | 0.33 |
| | Uitz et al. (2012) | Bloom region | May-Aug 1998-2007 | 0-1.5$Z_{eu}$ | 0.26–0.82 |
| | - | No blom region | - | - | 0.22–0.69 |
| Biogeochemical model | Lacroix & Nival (1998) | Ligurian Sea | | 0-200 m | 0.13 |
| | Allen et al. (2002) | Ligurian Sea | | 0-$Z_{eu}$ | 0.33 |
| | - | Ionian Sea | | - | 0.14 |
| In-situ $^{14}$C measurements | Minas (1970) | Northwestern basin | 1961-1965 | Surface | 0.21 |
| | Magazzu & Decembrini (1995) | Ionian Sea | 1983-1992 | 0-$Z_{eu}$ | 0.22 |
| | Turley et al. (2000) | Ligurian Sea | Oct 1997, Apr-May 1998 | 0-$Z_{eu}$ | 0.5 |
| | Marañon et al. (2021) | Ionian Sea | May 2017 | 0-200 m | 0.19 |

**Gross community production**

| Method | Reference | Area | Period | Layer | Estimate |
|---|---|---|---|---|---|
| $c_p$ diel cycle-based method | Barnes & Antoine (2014) | Ligurian Sea | May-Aug 2006-2011 | 0-$Z_{eu}$ | 0.8-1.5 |





**Table 5**: Comparison of the mean rates ± SD (gC m$^{-2}$ d$^{-1}$) of the community production integrated within the euphotic layer, derived from the application of the bio-optical diel cycle-based method to the $c_P$ measurements, using different bio-optical relationships from the literature for converting the $c_P$ values into POC biomass.

| Reference | Ligurian Sea | Ionian Sea |
|---|---|---|
| Marra et al. (1995) | 0.89±0.84 | 0.35±0.09 |
| Claustre et al. (1999) | 1.22±1.16 | 0.48±0.12 |
| Oubelkheir et al. (2005) | 1.18±1.13 | 0.46±0.11 |
| Behrenfeld & Boss (2006) | 1.43±1.35 | 0.56±0.14 |
| Gardner et al. (2006) | 0.93±0.88 | 0.36±0.09 |
| Stramski et al. (2008) | 1.62±1.54 | 0.63±0.16 |
| Loisel et al. (2011) | 0.98±0.92 | 0.38±0.10 |
| Cetinić et al. (2012) | 0.96±0.91 | 0.37±0.09 |





**Table 6**: Community production mean rates ± SD (gC m$^{-2}$ d$^{-1}$) derived from the application of the bio-optical diel cycle-based method to the $c_p$ measurements in the two considered regions. The production rates are integrated within the surface, subsurface maximum, and euphotic layers.

| Variable | Ligurian Sea | | | Ionian Sea | | |
|---|---|---|---|---|---|---|
| | Euphotic | Surface | SCM | Euphotic | Surface | SCM |
| POC (gC m$^{-2}$ d$^{-1}$) | 3.67±1.11 | 0.36±0.17 | 3.86 ± 1.20 | 1.88 ±0.24 | 0.34±0.14 | 0.93±0.31 |
| GCP (gC m$^{-2}$ d$^{-1}$) | 1.18±1.13 | 0.29±0.33 | 0.96±1.28 | 0.46±0.11 | 0.11±0.04 | 0.14± 0.39 |





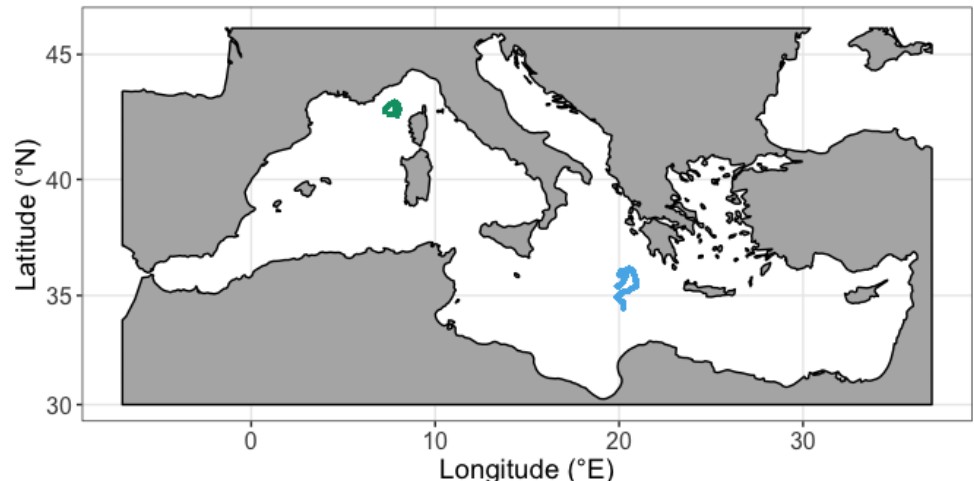

**Figure 1**: Trajectories of the two BGC-Argo profiling floats fLig (WMO6901776) and fIon (WMO6902828) deployed respectively in the Ligurian Sea (green) and the Ionian Sea (blue).

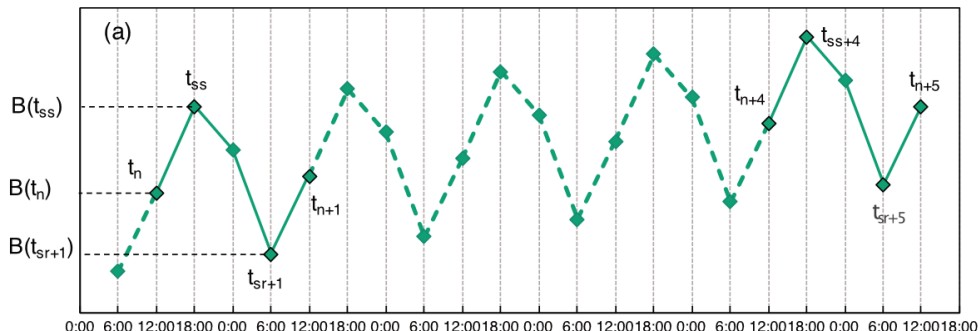

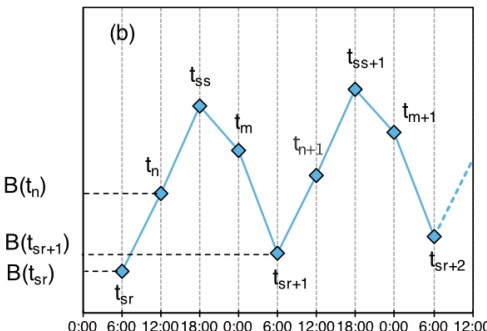

**Figure 2**: Schematic representation of the diel variations of the depth-integrated bio-optical properties converted to POC biomass (B) and the sampling strategies employed in the (a) Ligurian Sea and (b) Ionian Sea. The diamond-shaped symbols indicate schematically the float profile times, labeled with time stamps associated with sunrise (sr), noon (n), sunset (ss) and midnight (m), with the corresponding POC biomass estimated within the considered layer (e.g., $B(t_{sr})$, etc.). The numeric subscripts (+1, +2, +4 or +5) indicate the number of days since the first profile of the summertime time series.



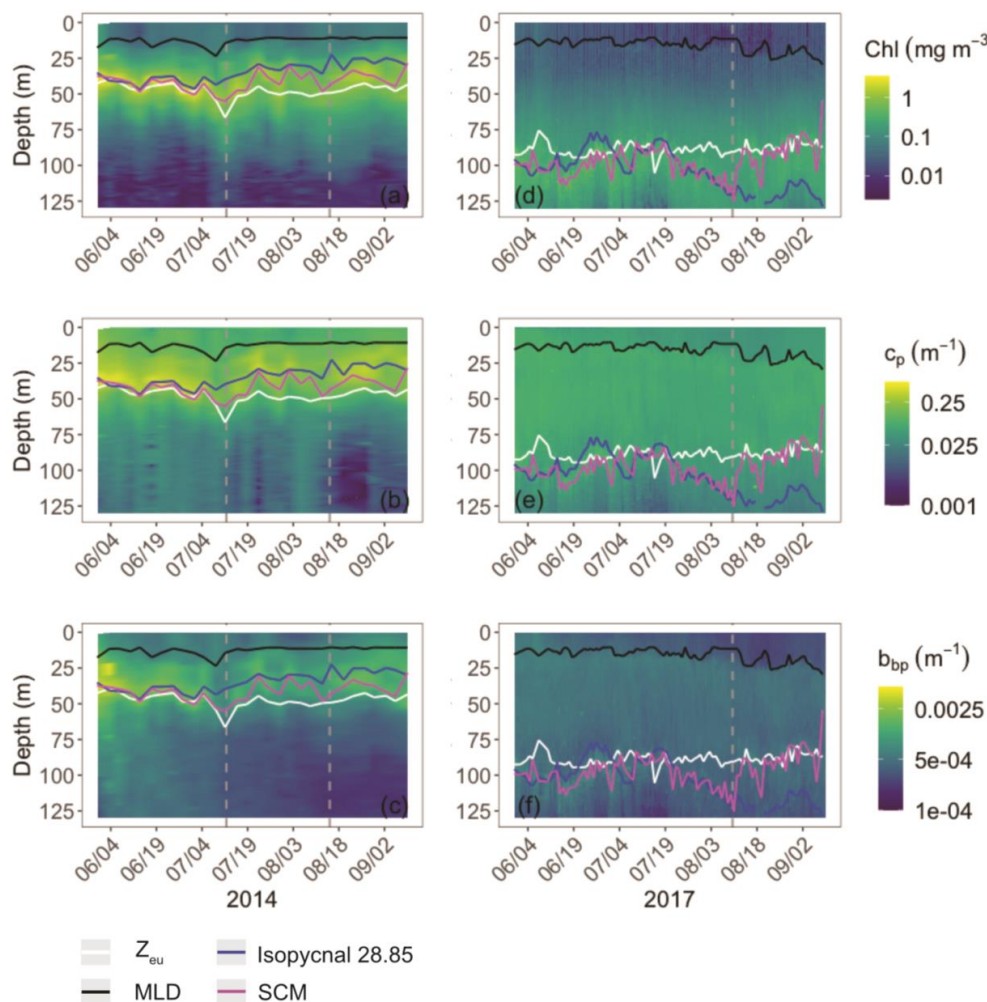

**Figure 3**: Time series of Chl (a and d), $b_{bp}$ (b and e) and $c_p$ (d and f) in the Ligurian Sea (left) and the Ionian Sea (right). The Mixed Layer Depth (MLD; black line), the isopycnal 28.85, expressed as $\sigma_t$ (blue line), the euphotic depth ($Z_{eu}$; white line) and the depth of the SCM (magenta line) are superimposed onto the bio-optical timeseries. The dashed lines indicate the dates at which the $c_p$ and the $b_{bp}$ values in the SCM layer reach a minimum.

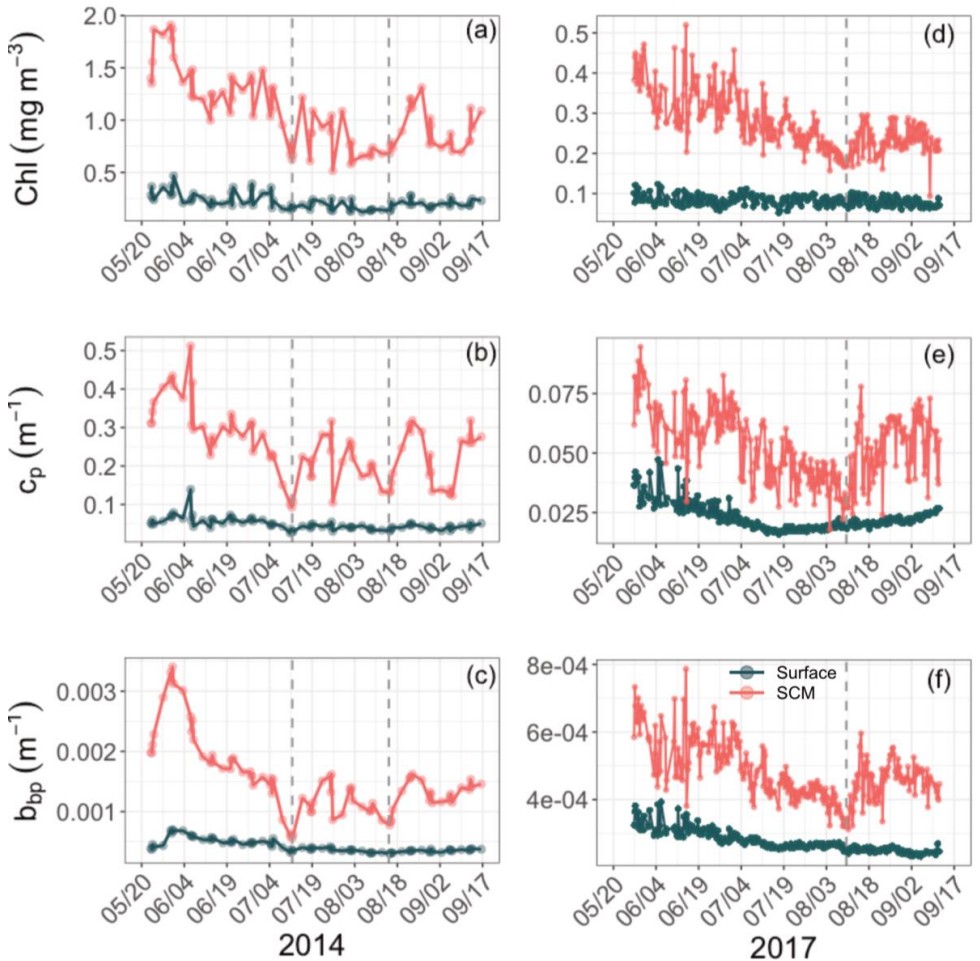

**Figure 4**: Temporal evolution of Chl (a and d), $c_{bp}$ (b and e), and $b_{bp}$ (c and f) in the surface (red) and SCM (dark green) layers for the Ligurian Sea (left) and the Ionian Sea (right). The dashed lines indicate the dates when the values of $c_p$ and $b_{bp}$ in the SCM layer reach a minimum.

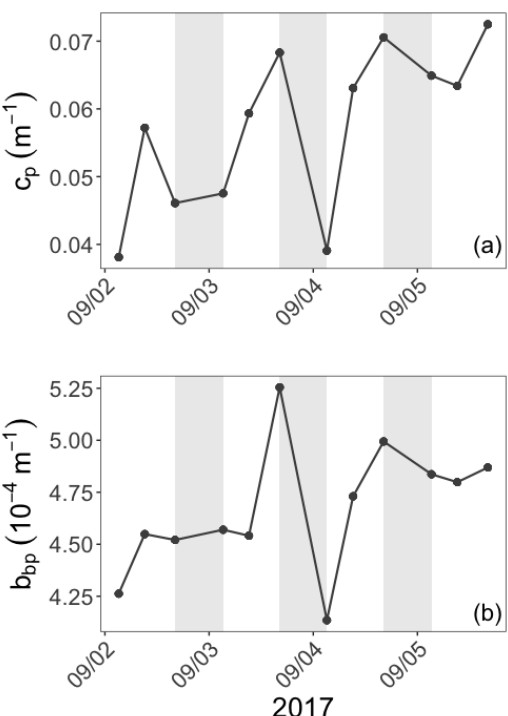

**Figure 5**: Example of the variations of the $c_p$ (a) and $b_{bp}$ (b) coefficients at the daily time scale in the Ionian Sea in the SCM layer during the interval from September 2 to September 6, 2017. The grey shaded area indicates the nighttime.





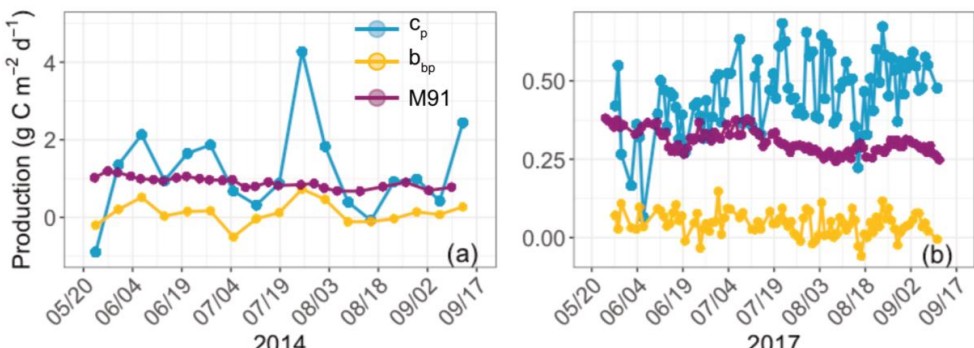

**Figure 6**: Comparison of the biological production integrated within the euphotic layer, derived from the diel cycle of $c_p$ (blue) or $b_{bp}$ (yellow) or computed using the bio-optical primary production model of Morel (1991) (purple) for the Ligurian Sea (a) and the Ionian Sea (b).

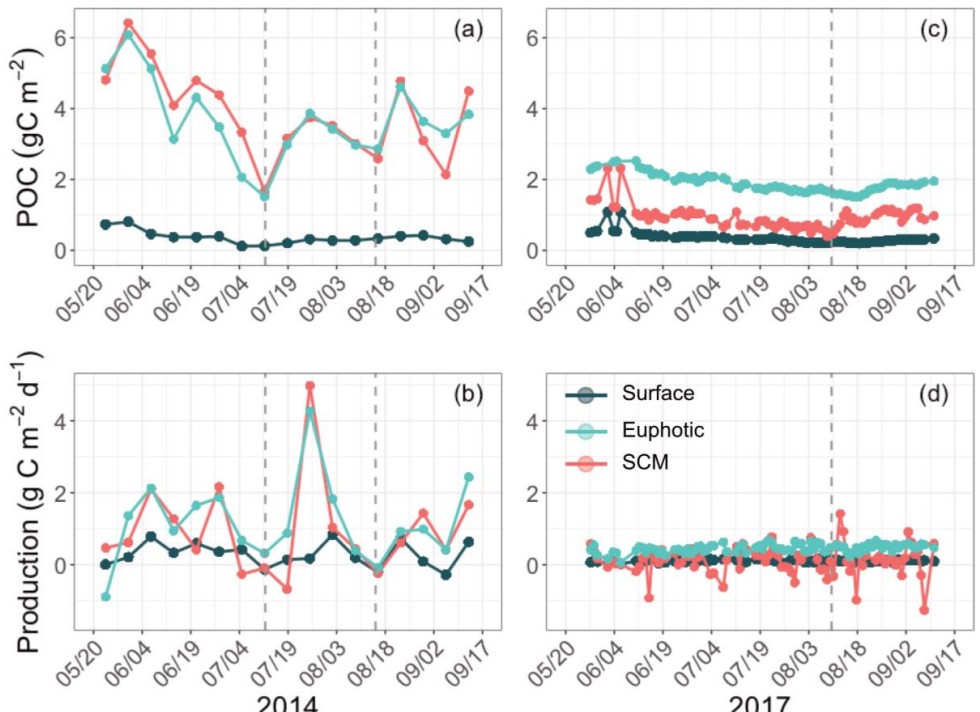

**Figure 7**: Temporal evolution of the POC and community production derived from the diel cycle of $c_p$ in the Ligurian Sea (a– b) and the Ionian Sea (c–d) and integrated in three different layers of the water column: surface (dark green), euphotic (light blue) and SCM (red) layers. The dotted lines indicate the dates when $c_p$ in the SCM layer reaches a minimum.



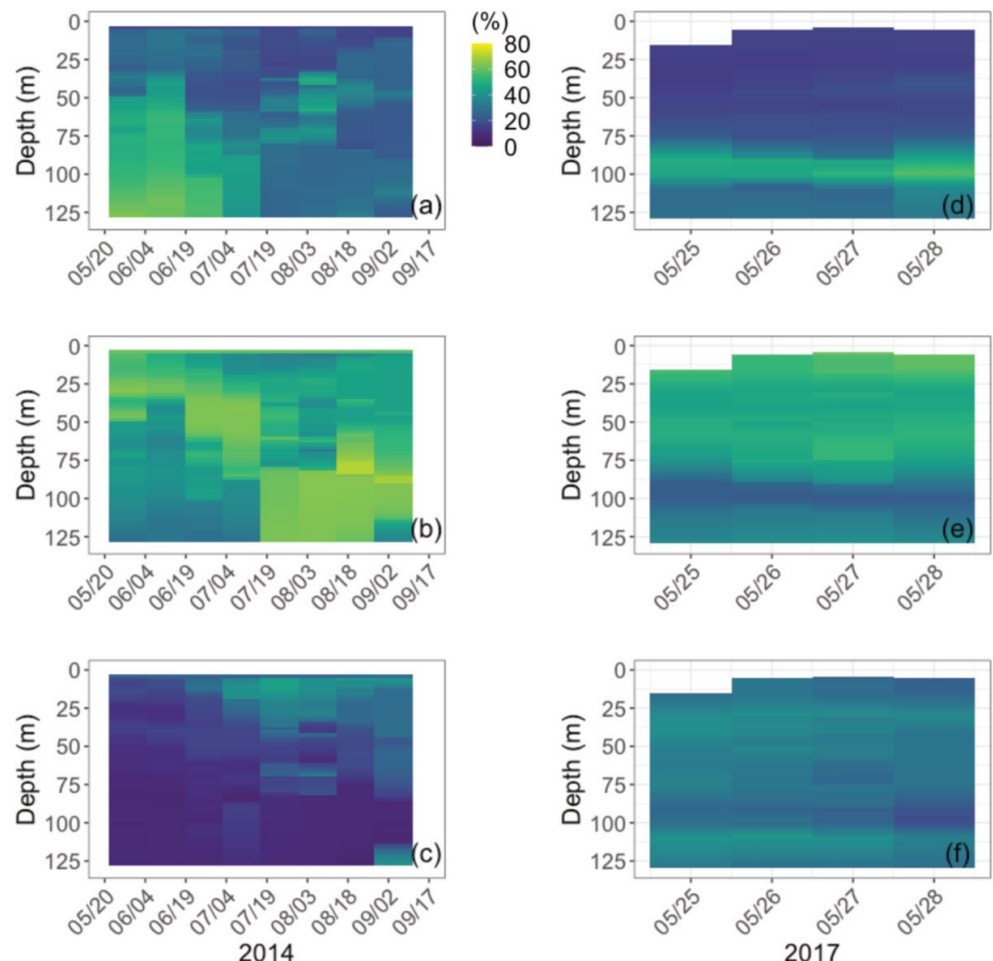

**Figure 8**: Depth-interpolated time series of the relative contributions to the chlorophyll *a* concentration (%) of the micro- (a and d), nano- (b and e) and picophytoplankton (c and h) derived from HPLC pigment determinations in the Ligurian Sea (BOUSSOLE site; left) and the Ionian Sea (PEACETIME cruise; right). The pigment data were collected at the BOUSSOLE site in the same region and at the same time period as the fLig float deployment on May 24, 2014 (see text section 2.1). The fION float was deployed concurrently to sampling for HPLC pigments at the PEACETIME ION station on May 28, 2017. Pigment data collected at ION over four days prior to float deployment are shown.





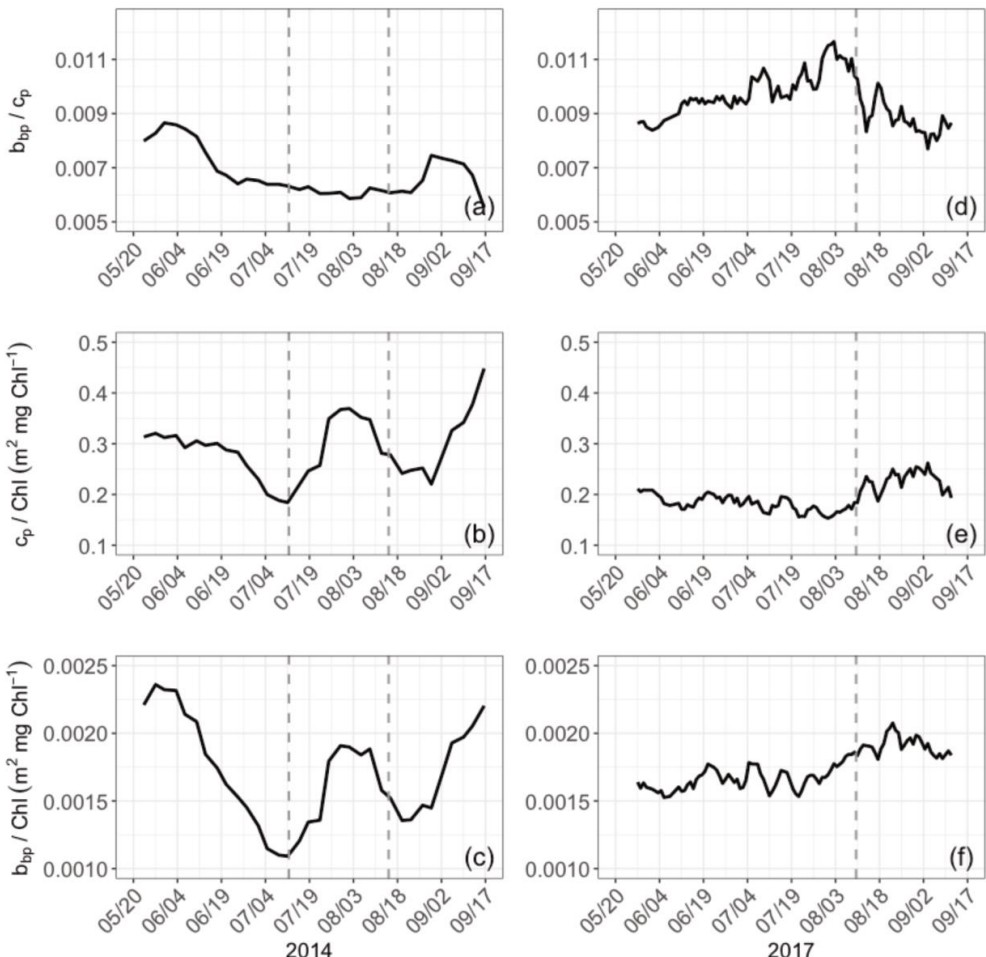

**Figure 9**: Temporal evolution of the bio-optical ratios of $b_{bp}$ / $c_p$ (a), $c_p$ / Chl (b) and $b_{bp}$ / Chl (c) in the SCM layer for the Ligurian Sea (left) and the Ionian Sea (right). The dotted lines indicate the dates when the values of $c_p$ in the SCM layer reach a minimum.





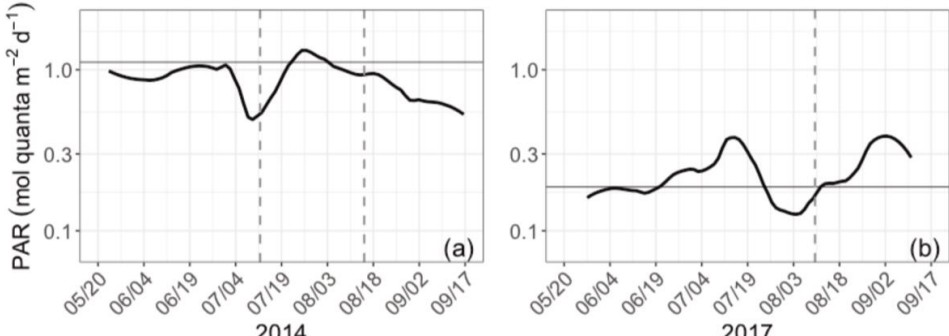

**Figure 10**: Time series of the daily-integrated photosynthetically available radiation (PAR) at the SCM level in the Ligurian Sea (a) and the Ionian Sea (b). The horizontal grey line shows the median of each time series. The dotted lines indicate the dates at which the values of $c_p$ in the SCM layer reach a minimum.