# Peer review of "Biological production in two contrasted regions of the Mediterranean Sea during the"

_Biogeosciences, 2021_

## Author Response (AR1)

Dear Co-Editor-in-Chief and Associate Editor,

We are grateful to both Reviewers for their detailed review and constructive comments. Our responses to each of their comments are provided in the attached documents. We also thank the Co-Editor-in-Chief for her questions which we address below.

We hope that we have provided satisfactory responses to the comments raised by the Reviewers, Associated Editor and Co-Editor-in-Chief, and made appropriate revisions in the new version of our manuscript which we provide in attachment.

We thank you for your time and consideration of our manuscript.

Sincerely,

Julia Uitz, on behalf of all co-authors

-Table 1: entry for Loisel et al. (2011): the equation parameter is Cp not Cbp correct?
Thank you. We have corrected the $c_p$ symbol in Table 1.

-Lines 359-360: It would be helpful to report the position where the HPLC samples were taken with respect to the buoy positions in Fig. 1.
Thank you for the suggestion. We have added a symbol showing the location of the BOUSSOLE site on the map in Fig. 1 (please see the revised figure below).

-Lines 376-380: if the plankton properties are considered similar, why not use also POC values taken during the BOUSSOLE and Peacetime cruises to check the Cp and Bbp derived POC?
POC is not part of the core parameters of the BOUSSOLE program. It has only been occasionally sampled and not simultaneously with the fLig float timeseries. Solely one POC profile measured at the deployment of the fIon float was available.

Our approach here has been similar for POC as for fluorescence-derived chlorophyll $a$. At this stage, it is not possible to account for the natural variability in the bio-optical ($c_p$- or $b_{bp}$-to-POC) relationships, occurring with depth and over the lifetime of the floats. Hence, we believe that presenting an analysis of the sensitivity of the computed production rates to selected relationships from the literature is more robust and informative as it enables to provide a range of production values for each of the considered regions.

-Fig.4.: Are these averages for the corresponding layers? The highlighted minima for the Ligurian sea seem a bit arbitrary (i.e. a minimum is also observed end of July)?
Fig. 4 shows average value of each variable within the surface and SCM layers. For the Ligurian Sea, we observe two local minima delineating a peak between mid-July and mid-August (in the revised manuscript, we added "local" to specify that these are not the only

minima in the time series, l. 555). This peak coincides with a *"moderate peak in POC"* (l. 680) and a large peak in production, *"associated with maximum production rates"* (l. 684). This is the reason why our analysis focuses on these two minima. We modified the text in order to clarify this point: *"These temporal patterns are further discussed in relation with the variability in the estimated POC and production rates (Section 3.4)"* (l. 564–565).

-Table 3 and Figure 4: Based on the temporal dynamics and values presented in Fig. 4 one would expect the values for the SCM in Table 3 to be the most variable, can you explain why this is not the case?

Table 3 shows the variability in the $c_p$ and $b_{bp}$ coefficients from one day relative to the next day, which, in most cases, is larger for the SCM than for the surface layer. Nevertheless, this variability may not be fully assessed visually from Fig. 4 due to the scale of the y-axis that matches the values measured within the SCM layer (not those measured in the surface layer). As an illustration, we provide a figure similar to Fig. 4, but only for the surface layer (please see Fig. R at the end of this document).

-Lines 468-469: Phytoplankton species often accumulate C during the day but divide at night (ex: Kottmeier etal., 2020,doi: 10.1002/lno.11418), could that be a reason for the weak day-night cycle in Bbp.

The present approach indeed relies on diel changes in $c_p$ or $b_{bp}$, with the daytime increase essentially reflecting photosynthetic cellular carbon accumulation, and the nighttime decrease attributed to cell division as well as respiration, aggregation, sinking, and grazing. This has been clarified in the revised version of the manuscript: *"The diurnal increase in $c_p$ or $b_{bp}$ has been primarily attributed to photosynthetic cellular organic carbon production (Siegel et al. 1998), that will first result in an increase in cell size, or an increase in cell abundance and a decrease in cell size following cell division often occurring at night. [...] The nighttime decrease in $c_p$ or $b_{bp}$ may be explained by a decrease in cellular abundance due to aggregation, sinking or grazing (Cullen et al. 1992), a reduction in cell size and/or refractive index associated with cell division and respiration."* (l. 349–357).

The $b_{bp}$ coefficient shows a weaker daily cycle than $c_p$ because these two coefficients respond differently to the size and composition of the particle pool. In particular, the contribution of phytoplankton to $c_p$ is more important than to $b_{bp}$ that is more sensitive to submicron (non-algal) particles, as explained in in l. 592–600: *"phytoplankton make a larger contribution to $c_p$ than $b_{bp}$, in part due to their strong absorption efficiency. In addition, $b_{bp}$ is more sensitive to smaller (<1 µm) particles (Stramski & Kiefer 1991; Ahn et al. 1992; Stramski et al. 2001; Boss et al. 2004) and to particle shape and internal structure (Bernard et al. 2009; Neukermans et al. 2012; Moutier et al. 2017; Organelli et al. 2018). While the diel cycle of $c_p$ would be essentially driven by photosynthetic processes due to the influence of phytoplankton on $c_p$, $b_{bp}$ would be more responsive to detritus and/or heterotrophic bacteria that show minor, if not negligible, daily variability. Hence, such specificities in the bio-optical coefficients may explain the observed differences in their diel cycles."*

Lines 539-542: see previous comment.

It is here referred to the following statement: *"The bio-optical diel cycle-based method produces several occurrences negative values in the SCM layer, indicating that the quasi-1-D assumption is occasionally not satisfied in the lower part of the euphotic layer. This could arise when physical processes that transport particles are larger than local growth and loss of POC"* (l. 539–542 in the original version of the manuscript). In the present approach, *"P is calculated as the sum of the net daily changes in POC biomass plus POC losses"* (l. 453–454). Hence negative values may arise if POC at sunset ($B_{ss}$) is lower than POC at sunrise the next day ($B_{sr+1}$), leading the loss term to be negative, which occurs when the 1D assumption is not satisfied (l. 703–705). This has been clarified in the methodology following a comment by Reviewer #1. We also refer the Co-Editor-in-Chief to our response to her previous question. Lower diel variability in $b_{bp}$ than in $c_p$ is caused by the different size and nature of the particle pool that impact differently these two coefficients, leading to different diel cycles.

-Fig. 8: It would be helpful to have the position of the MLD, SCM and Zeu reported in the Figures.
We have reported the respective position of the three layers on Fig. 8 (please see revised version of the figure below).

Finally, significant aspects in the discussion seem similar to results from the study by Maranõn et al. published in this issue and could be highlighted.
In addition to previous references to Marañón et al. (*"the SCM level is characterized by an enhanced contribution of microphytoplankton (diatoms) to the algal assemblage (49±5%; Fig. 8f), as discussed in Marañón et al. (2021)"* l. 733–735; *"our observations are consistent with previous studies reporting … the occurrence of diatoms at depth (Siokou-Frangou et al. 2010; Crombet et al. 2011; Marañón et al. 2021)"* l. 739–744), we have completed the manuscript in order to highlight further the consistency between Marañón's findings and those of our study: *"This result supports previous findings that indicate, contrary to the classic view of diatoms thriving essentially in dynamic eutrophic conditions, these organisms have the ability to maintain in stratified oligotrophic environments, including in deep layers under low light-nutrient conditions (Kemp & Villareal 2013; Kemp & Villareal, 2018). This was also highlighted by Marañón et al. (2021) based on observations in the Mediterranean Sea (PEACETIME cruise)."* (l. 860–865); *"Yet they suggest that the contribution of the SCM layer to the water column production may be significant globally, although commonly ignored. Our observations are consistent with previous findings in the Mediterranean Sea (Crombet et al. 2011; Marañón et al. 2021) and in other regions of the world ocean (Kemp & Villareal 2013; Mignot et al. 2014)."* (l. 888–891).

[Figure]

**Figure 1**: Trajectories of the two BGC-Argo profiling floats fLig (WMO6901776) and flon (WMO6902828) deployed respectively in the Ligurian Sea (green) and the Ionian Sea (blue) superimposed onto a 9-km resolution summer climatology of surface chlorophyll *a* concentration (in mg m$^{-3}$) derived from MODIS Aqua ocean color measurements. The asterisk-shaped symbol indicates the geographic location of the BOUSSOLE site.

[Figure]

**Figure R**: Temporal evolution of *Chl* (a and d), $c_p$ (b and e), and $b_{bp}$ (c and f) in the surface layer for the Ligurian Sea (left) and the Ionian Sea (right).

[Figure]

**Figure 8**: Depth-interpolated time series of the relative contributions to the chlorophyll *a* concentration (%) of the micro- (a and d), nano- (b and e) and picophytoplankton (c and h) derived from HPLC pigment determinations in the Ligurian Sea (BOUSSOLE site; left) and the Ionian Sea (PEACETIME cruise; right). The pigment data were collected at the BOUSSOLE site in the same region and at the same time period as the fLig float deployment (see text section 2.1). The fIon float was deployed concurrently to sampling for HPLC pigments at the PEACETIME ION station. Pigment data collected at ION over four days prior to float deployment are shown. As an indication, the depths of the euphotic depth ($Z_{eu}$; white dashed line), mixed layer (MLD; black dashed line) and SCM (magenta dashed line) derived from the BGC-Argo float measurements, as in Fig. 3, are overlaid onto the pigment data.

**Responses to Reviewer #1**

Manuscript # bg-2021-123 "Biological production in two contrasted regions of the Mediterranean Sea during the oligotrophic period: An estimate based on the diel cycle of optical properties measured by BGC-Argo profiling floats", by M. Barbieux, J. Uitz, A. Mignot, C. Roesler, H. Claustre, B. Gentili, V. Taillandier, F. D'Ortenzio, H. Loisel, A. Poteau, E. Leymarie, C. Penkerc'h, C. Schmechtig, and A. Bricaud.

We appreciate the constructive comments and suggestions from the Reviewer. Here we present our detailed responses to the Reviewer's comments as well as the changes made to the manuscript in order to address these comments. The Reviewer's comments are in black, our responses follow each comment in blue. The line numbers refer to the revised version of the manuscript in track change mode.

**GENERAL COMMENTS**

This study is an original contribution to the field of bio-optical oceanography. Though the approach used in this study is not new, the ms provide a detailed analysis of a unique dataset that includes high-resolution vertical measurements of biogeochemical properties acquired in two ecoregions of the Mediterranean Sea, during summer oligotrophy.

The application of the cp-based approach to estimate particles production from such data provides new findings, that significantly improve our understanding of particles dynamics in oligotrophic areas. The ms successfully addresses the challenge of filling typical observation gaps in a traditionally under-sampled ocean.

The introduction clearly introduces the general context and specific objectives of the present study.

The M&M is detailed, sound and robust. The structure of the M&M is sometimes a bit confusing, and there this some overlap between sections 2.3 – 2.5 and section 2.6. I have two major methodological comments (see below comment #2), and would suggest to display more examples cp and bbp measurements (see below comment #1).

The Results and Discussion, which are merged in a single section, are nicely structured and generally well supported.

The conclusion is too long, and there is some degree of redundancy with the discussion. Synthetizing the conclusion to the main results and "take-home messages" would improve the article.

We thank the Reviewer for his/her positive comments on our study. We address his/her questions about the methodology and structure of the manuscript point by point below.

**SPECIFIC COMMENTS**

1) As the whole study is built on cp and bbp measurements, it is necessary to show the data (an appendix could be used for that purpose). A schematic representation of the cp diel variability (Figure 2) is useful to explain the method, and the example provided in Fig. 5 (Ionian Sea) is interesting, but this is not sufficient.

The full $c_p$ and $b_{bp}$ dataset used for quantifying the diel variability in the optical properties and the derived POC production is shown for both the surface and SCM layers of the Ligurian and Ionian Seas in Fig. 4. In order to accommodate the Reviewer's comment, and following an approach similar to Gernez et al. (2011) and Kheireddine and Antoine (2014), we have added as appendix two figures shown at the end of this document and modified the text accordingly: "*Diel cycles, characterized by a daytime increase and a nighttime decrease, are observed in both $c_p$ and $b_{bp}$ time series in all three layers of the water column, as illustrated for the SCM layer of the Ionian Sea in Fig. 5 (examples of the diel cycles of $c_p$ and $b_{bp}$ for both the Ligurian and Ionian Seas are provided in Appendix A)*" (Sect. 3.2.1 l. 568–571).

2) Here, I raise two methodological comments concerning section 2.3. "Characterization of the diel cycle of the bio-optical properties".

First comment: in the literature, the diel variability is generally defined as the change in cp between sunrise and sunset (Siegel et al., 1989; Cullen et al., 1992, etc.). Such daytime increase in cp has been previously associated to particle growth and production.

In the present study, the diel variability is computed as the relative variation between two sunrises (in the Ionian Sea, eq. 3) or two noons (in the Ligurian Sea).

We thank the Reviewer for this important remark that allowed us to identify an error in the manuscript, not in the method but in its presentation. The description and the representation (Fig. 2) of the sampling scheme of the Ligurian float was actually erroneous. The time reference used for computing diurnal changes in $c_p$ and $b_{bp}$, further converted into production, is actually identical for both the Ligurian (fLig) and Ionian (fIon) floats, i.e. sunrise to sunset.

In the revised version of the manuscript, we have corrected Fig. 2 (see new version below) and modified the accompanying text as follows: "*The fLig float cycle commences with the first profile at sunrise ($t_{sr}$), a second at solar noon ($t_n$), a third profile at sunset the same day ($t_{ss}$), and a fourth*

*profile at sunrise the next day ($t_{sr+1}$). The fLig float then acquires a profile at solar noon 4 days later ($t_{n+4}$), and then restarts 3 days later the acquisition of 4 profiles in 24 hours from sunrise ($t_{sr+7}$)."* (Sect. 2.2 l. 295–299).

We have also corrected the text in Sect. 2.3: *"We also consider the relative daily variation $\widetilde{\Delta c_p}$ and $\widetilde{\Delta b_{bp}}$ (expressed as % change) for each float and each day of observation, from sunrise to sunrise ... with $c_p(t_{sr})$ and $b_{bp}(t_{sr})$ being the values of $c_p$ and $b_{bp}$ at sunrise and $c_p(t_{sr+1})$ and $b_{bp}(t_{sr+1})$ the values at sunrise the next day"* (l. 318–323).

Did the authors also characterize the sunrise-to-sunset variability?

We used the indicators of Gernez et al. (2011) and Kheireddine & Antoine (2014) to characterize the relative daily variations (sunrise to sunrise the next day) of the $c_p$ and $b_{bp}$ coefficients. Following the Reviewer's question, we have characterized the amplitude of the $c_p$ and $b_{bp}$ diurnal variations (sunrise to sunset) and modified Sect. 2.3 as follows:

*"... we use the metrics defined by Gernez et al. (2011) and Kheireddine & Antoine (2014). First, we compute the amplitude of the diurnal variation of the $c_p$ and $b_{bp}$ coefficients as:*

$$\Delta c_p = c_p(t_{ss}) - c_p(t_{sr}) \tag{3a}$$

$$\Delta b_{bp} = b_{bp}(t_{ss}) - b_{bp}(t_{sr}) \tag{3b}$$

*with $c_p(t_{sr})$ and $b_{bp}(t_{sr})$ the values of $c_p$ and $b_{bp}$ at sunrise and $c_p(t_{ss})$ and $b_{bp}(t_{ss})$ the values at sunset the same day"* (Sect. 2.3 l. 311–317).

We have also added to Sect. 3.2.1 the following text: *"Considering the time series of the Ligurian and Ionian Seas, as well as the surface and SCM layers, the $c_p$ and $b_{bp}$ coefficients show mean diurnal amplitudes, $\Delta c_p$ and $\Delta b_{bp}$, spanning between 0.001 $m^{-1}$ and 0.02 $m^{-1}$ and 7 x $10^{-6}$ $m^{-1}$ and 9 x $10^{-5}$ $m^{-1}$, respectively. These results are consistent with Gernez et al. (2011), who observed $\Delta c_p$ values ranging within 0.01 $m^{-1}$ and 0.07 $m^{-1}$ in the surface layer of the Ligurian Sea (BOUSSOLE mooring) during the summer to fall oligotrophic period. Relative to the mean $c_p$ and $b_{bp}$ values, the mean $\Delta c_p$ and $\Delta b_{bp}$ correspond to diurnal variations of 9–20% and 5–10%, respectively."* (l. 571–578).

Second comment: the time reference used to characterize the daily changes in cp are not the same in the Ionian Sea (reference = sunrise) and in the Ligurian Sea (reference = noon). This introduces a bias in the comparison between the results from both study sites, and should be discussed.

Would it be possible to compute the diel variability using the same time reference (e.g. sunrise) in both cases? The same two comments also apply to Section 2.6.3 "Calculation of the production rate".

As indicated in our response to the Reviewer's previous comment, the sampling scheme of the fLig float was incorrectly described in the manuscript. Both the fLig and fIon measurements enable to characterize daily variations in $c_p$ and $b_{bp}$ from sunrise ($t_{sr}$) to sunrise the next day ($t_{sr+1}$). Hence no bias is introduced in our estimates. This has been corrected in the revised version of the manuscript.

In eqs. 9-12, the authors explain how a daily and depth-integrated production of particles, P can be inferred from diel changes in cp. Is the variable "P" a proxy of the net community production (NCP) as defined in Claustre et al 2008? Or is it something different? It would be useful to add this precision in subsection 2.6.3. It is not clear whether the production has been inferred using the day-to-day or the daytime (sunrise to sunset) change in the cp-derived POC.

How was the gross community production (GCP) estimated?

We apologize for any lack of clarity in Sect. 2.6.3. The variable P designates gross community production, resulting from particle growth over 24h, i.e., from sunrise ($t_{sr}$) to sunrise the next day ($t_{sr+1}$), and within the layer of the water column comprised between depths z1 and z2:

$$"P = \int_{t_{sr}}^{t_{sr+1}} \int_{z2}^{z1} \mu(z,t)\, b(z,t)\, dz\, dt, \qquad\qquad (9)" (l.447)$$

In practice, it is calculated as the depth-integrated net variation of POC over 24h plus POC losses:

$$"P = B_{t_{sr+1}} - B_{t_{sr}} + l \int_{t_{sr}}^{t_{sr+1}} B(t)\, dt. \qquad\qquad (11)" (l.452)$$

We have clarified this point in the revised version of the manuscript through the following modifications:

"The daily (24-hour) depth-integrated gross production of POC, P (in units of gC m$^{-2}$ d$^{-1}$)" (l. 445).

"where the gross production P is calculated as the sum of the net daily changes in POC biomass plus POC losses, assuming a constant rate (l) during daytime and nighttime" (l. 453–454).

Here also, the time reference differs between the Ligurian and Ionian Sea (L343-345). How is it expected to influence the results? Would it be possible to standardize the method?

As indicated above, the daily variations in $c_p$ and $b_{bp}$ are characterized from sunrise ($t_{sr}$) to sunrise the next day ($t_{sr+1}$) for both the fLig and fIon floats.

3) As the instruments deployed at BOUSSOLE provide high-frequency measurements, it would be very interesting to compare the surface cp time-series acquired by the fLig float with the BOUSSOLE data (accounting the lag of 2 days identified in L371-380). Such a comparison would be useful to assess if some information is missed by the lower (but still high) temporal resolution of the BGC Argo float, as well as to assess the spatial representativity of BOUSSOLE point-based measurements.

We recognize that comparison of high frequency observations from the BOUSSOLE mooring with those acquired by the BGC-Argo float, covering a larger spatial scale, would be of interest. However, we believe that this would require a specific study that is beyond the scope of the present work that focuses on a comparison of two contrasted oligotrophic systems with a zoom on the subsurface layer.

**TECHNICAL COMMENTS**

In the title, the term "optical properties" is maybe to general, and could be more detailed (e.g. beam attenuation coefficient).

Although in Sect. 3.2.2 we disregard $b_{bp}$-derived production to focus only on $c_p$-derived production, our study also considers (and compares) $b_{bp}$ and $c_p$. Thus, we believe it would not be totally correct to mention only "beam attenuation coefficient" in the title, but it would also be too long to indicate both "particulate beam attenuation and backscattering coefficients". Unless it is a mandatory request from the Reviewer, we would prefer to keep the title as is.

The variables cp, bbp, Zeu, Zpd, etc. should be italicized throughout the ms

We have italicized the variables in the revised version in the manuscript.

L167-175 how was the time period of the two floats selected?

The aim of the study is to compare two different oligotrophic systems. Based on previous knowledge, the Ligurian Sea is oligotrophic in summer while the Ionian Sea is permanently oligotrophic. We thus selected the oligotrophic summer period for both the fLig and fIon floats. We used the entire time series of the fIon float and restricted the time series of the fLig float so it coincides in months with that of fIon. We have modified the text in order to clarify this point: "*The Ligurian Sea float (hereafter noted fLig, WMO: 6901776), was deployed in the vicinity of the*

*BOUSSOLE fixed mooring (7°54'E, 43°22'N) during one of the monthly cruises of the BOUSSOLE program (Antoine et al. 2008) and profiled from April 9, 2014 to March 15, 2015. For the purpose of this study focusing on oligotrophic systems, we selected the fLig float measurements acquired during the time period May 24 to September 13, 2014 to coincide in months with the Ionian Sea float time series.*" (Sect. 2.2 l. 199–204).

L265 typo in "Loisel et al."

We have corrected "Loisel e al. 2011" to "*Loisel et al. 2011*" (l. 342).

L287-300 and Table 1. The same cp-to-POC relationship is used in the Ligurian and Ionian Sea, despite the bio-optical differences between the two basins. Two different relationships have also been reported in the literature (Oubelkheir et al 2005, Loisel et al 2011). The authors decided to apply the results from Oubelkheir et al. (2005). Why not from Loisel et al (2011)? This could be added to the discussion in L502-508.

Several bio-optical relationships, linking $c_p$-to-POC or $b_{bp}$-to-POC, were considered in our analysis. The production estimates derived from these relationships, including that of Loisel et al. (2011), are provided in Table 5.

The justification of our choice is presented in Sect. 2.5: "*In the present study, we used the relationships from Oubelkheir et al. (2005) and Loisel et al. (2011) for $c_p$ and $b_{bp}$, respectively. Both relationships were established from in situ measurements collected in the Mediterranean Sea and produce $c_p$- or $b_{bp}$-derived POC values falling in the middle of the range of all the POC values resulting from the different bio-optical relationships taken from the literature (Tables 1 and 2)*" (Sect. 2.5 l. 401–407).

We also note that the variability in the production estimates induced by the choice of the bio-optical relationships is discussed in Sect. 3.2.2 (*l. 649–661*): "*The empirical relationships linking the $c_p$ (or $b_{bp}$) coefficient to POC are known to exhibit regional and seasonal variability … For the time period and study regions here, the $c_p$-based community production varies by a factor of 2, depending on the selected bio-optical relationship… Compared to the reference value obtained using the Oubelkheir et al. (2005) relationship, the $c_p$-based estimates are 25% lower and 37% higher using the relationships of Marra et al. (1995) and Stramski et al. (2008), respectively.*"

In order to consider specifically the relationship of Loisel et al. (2011), we have added a sentence indicating that: "*using the Mediterranean relationship of Loisel et al. (2011), instead of that of Oubelkheir et al. (2005), would reduce the $c_p$-based estimates by 17% in both study regions (Table 5). That said, although the absolute magnitudes vary depending upon proxy choice, the differences observed between locations is robust.*" (l. 661–666).

Figure1: it would be useful to display the summertime-averaged satellite-derived Chl concentration in Figure 1

We have added as background a summer climatology of ocean color-derived surface chlorophyll *a* concentration (see revised Fig. 1 below).

Figure 2: adding grey vertical bar to represent nighttime would help reading the graph

We have added grey shaded areas to indicate nighttime in the revised version of Fig. 2 (see below).

Figure 3: I suggest to add "Ligurian Sea" and "Ionian Sea" at the top of the left and right panels, respectively (and same comment for Figures 4, 6 – 10).

We have added the indication "Ligurian Sea" and "Ionian Sea" at the top of Figs. 2–4 and Figs. 6–10 as well as on the figures shown in Appendices A and B.

In Figure 3, displaying the cp-to-Chl ratio would also be useful to help reading section 3.1 (in particular L389 - 408)

In the previous step of the review process, we proposed, to accommodate the Reviewer's comment, to add two panels showing the time series of the $c_p$ / Chl ratio in the water column in the Ligurian and Ionian Seas along with a short following text in Sect. 3.1: "*In the Ligurian Sea, the SCM is intense … This induces vertical variations in the $c_p$-to-Chl ratio, with moderate values at surface, diminution at the level of the SCM, and maximum values below the SCM. In contrast, in the Ionian Sea, … the $c_p$-to-Chl ratio shows larger values in the upper part of the water column coinciding with larger values of the $c_p$ coefficient, and minimal values at the SCM due to a pronounced increase in Chl simultaneously to a reduction of $c_p$.*"

Nevertheless, we prefer not to insist on the distribution of the $c_p$-to-Chl ratio in Sect. 3.1 that aims to provide an overview of the biogeochemical and bio-optical characteristics, and keep it for Sect. 3.4.1 that largely presents and discusses the distribution of the bio-optical ratios. Discussing the $c_p$-to-Chl ratio requires some context and references, which are presented in Sect. 3.4.1 and would not be appropriate in Sect. 3.1. Hence, we rather not incorporate these modifications.

L482-485 the results also compare with the "delta POC" estimates reported in Gernez et al. (2011; see their fig. 14)

Here the Reviewer refers to the following sentence: *"The present $c_p$-derived values also compare favorably with … (0.8–1.5 gC $m^{-2}$ $d^{-1}$ in May–August; Barnes & Antoine 2014)"*, which we have modified accordingly: *"The present $c_p$-derived values also compare favorably with … (0.5–0.8 gC $m^{-2}$ $d^{-1}$ in Gernez et al. 2011; 0.8–1.5 gC $m^{-2}$ $d^{-1}$ in Barnes & Antoine 2014)"* (Sect. 3.2.2 l. 645–646)

L539-542 Here the authors assumed that negative values could be associated with particles transport. Please note that negative values could also occur if the losses exceed particles growth (which could occur if the community is dominated by heterotrophs).

Here we present estimates of gross community production, calculated as the sum of the net changes in POC over 24 h (from sunrise, $t_{sr}$, to sunrise of the next day, $t_{sr+1}$) plus POC losses. Therefore, negative values may arise if POC at sunset ($B_{ss}$) is lower than POC at sunrise the next day ($B_{sr+1}$), leading the loss term to be negative (Eq. 8). A negative loss term, which is biologically impossible, occurs if the 1D assumption is not satisfied. We hope that the clarification made to the methodological section will make this point clearer (see our response to the Reviewer's second comment).

L543-549 this is very interesting. It would be useful to provide the averaged cp/Chl (at the depth of the SCM) of the Ligurian Sea and Ionian Sea. Besides photo-acclimation, are there other ecophysiological hypothesis that would be consistent with this hypothesis? (e.g. composition and size distribution of the community of living particles, higher influence of nutrient stress in the Ionian Sea?).

Although, at first order, low values of the $c_p$ / Chl (or $b_{bp}$ / Chl) ratio are interpreted as resulting from photoacclimation, we acknowledge that several other sources of variations may be at play, as indicated in the following section (Sect. 3.4.1 l. 754–759): *"The $b_{bp}$ / Chl and $c_p$ / Chl ratios are both proxies for the POC / Chl ratio …, and thus an indicator of the contribution of phytoplankton*

*to the whole organic carbon pool. The variations are also interpreted as changes in the composition of phytoplankton communities … and their acclimation to the light-nutrient regime".*

Further, we mention, in addition to photoacclimation, different factors potentially responsible for low values of the bio-optical ratios: "*may reflect photoacclimation, by which phytoplankton organisms increase their intracellular Chl, and/or an increase in the fluorescence-to-Chl ratio in relation to limited or null non-photochemical chlorophyll fluorescence quenching… the invariant low $c_p$ / Chl and $b_{bp}$ / Chl values are consistent with both photoacclimation of phytoplankton to low-light conditions and a diatom-dominated phytoplankton assemblage (Cetinić et al. 2015; Barbieux et al. 2018). The relatively stable ratios observed in this region suggest a relative steadiness in the composition of the phytoplankton assemblage over the considered period.*" (l. 817–826).

L607-623 when discussing the range of variability of cp/Chl, please consider referring to Loisel & Morel (1998)

We thank the Reviewer for noting this oversight. We will add a reference to Loisel and Morel (1998) in the following sentences: "*The variations are also interpreted as changes in the composition of phytoplankton communities (e.g. Sathyendranath et al. 2009) and their acclimation to the light-nutrient regime (e.g. Geider et al. 1987; Loisel & Morel 1998; Geider et al. 1997; Cloern 1999)*" (Sect. 3.4.1 758–760); "*Low-light conditions typically prevailing in the SCM layer are usually associated with low values of the $c_p$ / Chl and $b_{bp}$ / Chl ratios (e.g. Loisel & Morel 1998; Behrenfeld & Boss 2003; Westberry et al., 2008; Barbieux et al. 2019)*" (l. 816).

We have also indicated that: "*These results are consistent with the study of Loisel & Morel (1998), reporting low values ranging within 0.1–0.2 $m^2$ mg $Chl^{-1}$ at the deep chlorophyll maximum level of oligotrophic sites*" (Sect. 3.4.1. l.783 –785).

L653 "it appears to result from changes in light conditions": light and/or nutrients?

The paragraph which the Reviewer refers to indicates that both light and nutrient availability plays a role in the observed production increase: "*the observed production episode may result from physical forcing that induces an upwelling of the water mass, thereby resulting in an alleviation of the light/nutrient limitation and an adequate balance between light and nutrient availability in the SCM layer*" (Sect. 3.4.2 l. 837–840).Then, in the next sentence, we do not question the role of nutrients, but simply point to the fact that changes in the light regime may explain photosynthetic (not bacterial) growth: "*Because it appears to result from changes in light conditions, we may*

*attribute this production event to phytoplankton (not community) growth*" (l. 843–845). We will replace "phytoplankton" by "photosynthetic" to clarify.

L679-689 as acknowledged by the authors, these are very hypothetical statements that are not supported by the present study. I therefore suggest to remove the quantitative results of such crude estimations from the conclusion (i.e. remove or re-write L745-747).

*As recommended by the Reviewer, we removed the sentence "More generally, our study suggests that the contribution of the SCM layer to the water column production varies broadly depending the considered system, whether seasonally (~42% in the Ligurian Sea) or permanently (~16% in the Ionian Sea) oligotrophic."* (l. 937).

*In order to make the conclusion more synthetic, as recommended by the Reviewer, we have kept only the essential results and the elements of recommendations or perspectives. This reduced the conclusion by about a third of its initial length.*

[Figure]

**Figure 1**: Trajectories of the two BGC-Argo profiling floats fLig (WMO6901776) and fIon (WMO6902828) deployed respectively in the Ligurian Sea (green) and the Ionian Sea (blue), superimposed onto a 9-km resolution summer climatology of surface chlorophyll *a* concentration (in mg m$^{-3}$) derived from MODIS Aqua ocean color measurements. The asterisk-shaped symbol indicates the geographic location of the BOUSSOLE site.

[Figure]

[Figure]

**Figure 2**: Schematic representation of the diel variations of the depth-integrated bio-optical properties converted to POC biomass (*B*) and the sampling strategies employed in the (a) Ligurian Sea and (b) Ionian Sea. The diamond-shaped symbols indicate schematically the float profile times, labeled with time stamps associated with sunrise (sr), noon (n), sunset (ss) and midnight (m), with the corresponding POC biomass estimated within the considered layer (e.g., $B(t_{sr})$, etc.). The numeric subscripts (+1, +2, +4, +7 or +8) indicate the number of days since the first profile of the summertime time series.

[Figure]

**Figure A1**: Example of time series of the $c_p$ coefficient in the surface (red) and SCM (dark green) layers, chosen within the time periods indicated by the dashed lines in Figs 3-4, from May 24 to July 14, 2014 (a), July 14 to August 16, 2014 (b), and August 16 to September 13, 2014 for the Ligurian Sea (left), and from May 28 to August 11, 2017 (d) and August 11 to September 11, 2017 (e) for the Ionian Sea (right).

[Figure]

**Figure A2**: Example of time series of the $b_{bp}$ coefficient in the surface (red) and SCM (dark green) layers, chosen within the time periods indicated by the dashed lines in Figs 3-4, from May 24 to July 14, 2014 (a), July 14 to August 16, 2014 (b), and August 16 to September 13, 2014 for the Ligurian Sea (left), and from May 28 to August 11, 2017 (d) and August 11 to September 11, 2017 (e) for the Ionian Sea (right).

**Responses to Reviewer #2**

Manuscript # bg-2021-123 "Biological production in two contrasted regions of the Mediterranean Sea during the oligotrophic period: An estimate based on the diel cycle of optical properties measured by BGC-Argo profiling floats", by M. Barbieux, J. Uitz, A. Mignot, C. Roesler, H. Claustre, B. Gentili, V. Taillandier, F. D'Ortenzio, H. Loisel, A. Poteau, E. Leymarie, C. Penkerc'h, C. Schmechtig, and A. Bricaud.

We appreciate the constructive comments and suggestions from the Reviewer. Here we present our detailed responses to the Reviewer's comments as well as the changes made to the manuscript in order to address these comments. The Reviewer's comments are in black, our responses follow each comment in blue. The line numbers refer to the revised version of the manuscript in track change mode.

**GENERAL COMMENTS**

The study aims to derive community production estimation from a dataset obtained by two BioGeoChemical-Argo (BGC-Argo) profiling floats deployed in the Ligurian and Ionian Seas. The authors mainly used the infradiel variability of cp and bbp measurements along the water column to investigate the diel variations of the bio-optical properties and thus of the production (estimated in particulate organic carbon). The authors compared the results between two contrasted areas of the Mediterranean Sea.

The study is well-done and convincing. The draft is well-written, well-structured and organized, starting an informative and well-documented introduction.

**Below some more specific comments**

**ABSTRACT**

Lines 18-19: "…marine biological production of organic carbon" … I suggest to be more precise: organic carbon is referred to particulate O.C.?; marine biological production is referred to phytoplankton, bacteria, zooplankton?

We have clarified the text as follows: "*This study assesses marine community production based on the diel variability*…" (l. 18).

Line 30: SCM layer (16-42%): I imagine that it is also (mainly?) dependent on the depth, i.e. on light availability?

We cannot expand the explanation in the abstract but, yes, this contribution varies depending in the type of system. We have made modifications in the aim to clarify this point: *"substantial contribution to the water column production of the SCM layer (16–42%), that largely varies with the considered system. In the Ligurian, the SCM…"* (Sect. 1 l. 29–30).

Lines 32-33: "the SCM is permanent induced by phytoplankton photoacclimation…" What does it mean?: does this SCM only the result of physiological acclimation/regulation of microalgae (increase of chl.a per cell)? Or/and to the fact that microalgae actively accumulate to this depth for many reasons (light, nutrient, stratification,…)

We have moderated our statement in the abstract (*"the SCM is permanent, primarily induced by…"* l. 32–33). This point is further discussed in response to another question from the Reviewer (please see below).

INTRODUCTION

Line 40: "Primary production is an essential component…" instead of component, process? Flux?

We have corrected as follows: "*Primary production is an essential process*" (l. 44).

Lines 57-60: what about active fluorescence measurements (FRRF?, etc.)

As suggested by the Reviewer, we have added citation of the FRRF technique in the Introduction section: *"Active chlorophyll fluorescence techniques, such as Fast Repetition Rate Fluorometry (FRRF), yield in situ phytoplankton physiological parameters, which when combined with appropriate modeling, provide estimates of derive primary production (e.g. Kolber & Falkowski 1993; Smyth et al. 2004). This technique has the major advantage of providing an instantaneous, fine-scale estimation of primary production in a non-invasive manner. Nevertheless, it is subject to assumptions and uncertainties, in particular related to the interpretation of fluorescence-light curve information in terms of carbon fixation, that still limit its use (see., e.g., Suggett et al. 2004; Corno et al. 2005; Regaudie-de-Gioux et al. 2014 and references herein)."* (l. 65–75).

Corno, G., Letelier, R.M., Abbott, M. R., and Karl, D.M.: Assessing primary production variability in the North Pacific Subtropical Gyre: A comparison of Fast Repetition Rate Fluorometry and $^{14}$C measurements, *J. Phycol.*, 42, https://doi.org/10.1111/j.1529-8817.2006.00163.x, 2005.

Kolber, Z. S., and Falkowski, P. G.: Use of active fluorescence to estimate phytoplankton photosynthesis in-situ, *Limnol. Oceanogr.*, 38, 1646–1665, 1993.

Regaudie-de-Gioux, A., Lasternas, S., Agustí, S., and Duarte, C. M.: Comparing marine primary production estimates through different methods and development of conversion equations, *Frontiers*, 1, https://doi.org/10.3389/fmars.2014.00019, 2014.

Smyth, T. J., Pemberton, K. L. , Aiken, J., and Geider, R. J.: A methodology to determine primary production and phytoplankton photosynthetic parameters from Fast Repetition Rate Fluorometry, *J. Plank. Res.*, 26, 11, 1337–1350, https://doi.org/10.1093/plankt/fbh124, 2004.

Suggett, D. J., Macintyre, H. L., and Geider, R. J.: Evaluation of biophysical and optical determinations of light absorption by photosystem II in phytoplankton, *Limnol. Oceanogr. Methods*, 316–332, https://doi.org/10.4319/lom.2004.2.316, 2004.

Line 99: "…and found weak results for the diel bbp cycle…" in which sense?, rephrase/explain

We have reworded this sentence as follows: "*and found that the diel cycle of $b_{bp}$ may not necessarily be interchanged with that of $c_p$, which calls for further investigations*" (l. 114–115).

Line 136-137: "…with an SCM induced mostly by photoacclimation (e.g., Mignot et al. 2014; Barbieux et al. 2019)". Please explain?

We have attemped to clarify this point, first, in the Introduction at the first occurrence of the term "photoacclimation": "*SCMs … are typically attributed to phytoplankton photoacclimation, the physiological process by which phytoplankton cells adjust to light limitation by increasing their intracellular chlorophyll content without concomitant increase in carbon (Kiefer et al. 1976; Cullen 1982; Fennel & Boss 2003; Letelier et al., 2004; Dubinsky & Stambler 2009).*" (Sect. 1 l. 137–140). Second, we have modified the sentence the Reviewer refers to as follows: "*with an SCM induced mostly by photoacclimation of phytoplankton cells without concomitant increase of carbon biomass*" (Sect. 1 l. 165–166).

METHODS
Lines 167-175: move to the section 2.2. (BGC-floats)

As suggested by the Reviewer, we have moved the sentence "*We deployed BGC-Argo floats programmed for "multi-profile" sampling in each of these two regions (Fig. 1)… the data sets arise from similar seasonal contexts*" to the beginning of Sect. 2.2. (l. 198–222).

Lines 196-201: "fluorescence-to-Chl ratio". If I understand well, the authors applied two successive types of correction: one related to NPQ and one on the differences between chl.a fluo and chl.a concentration. The last is considered basically from the chl.a fluo/chl.aratio = 2, applied on the two contrasted systems Ligurian and Ionian seas. I was wondering if using the same value for the two contrasted sites can represent an "error" for the interpretation of the results. The environmental (light, nutrients) and biological properties (microalgal communities) of the two sites are strongly different, that probably might affect the ratio fluo/chl.a.

The authors reported that they measured chl.a concentration with HPLC during the cruises they set-up for the deployment of the floats (lines 365-370). Can the authors use these data to retrieve a more precise chl.a fluo/chl.a ratio for the two sites?

We applied the standard BGC-Argo procedure consisting in the application of: the manufacturer (WETLabs) calibration coefficients; the NPQ correction of Xing et al. (2012); and the factor of 2 according to Roesler et al. (2017). Please note that the fluorescence-derived chlorophyll data distributed from the Coriolis Data Center are already corrected from the factor of 2 (as indicated in Sect. 2.2 l. 250–252: "*Hence the bias correction factor of 2 was applied to BGC-Argo fluorescence data … consistently with the processing performed at the Coriolis Data Center*").

Previous knowledge of the ratio of the fluorescence-derived Chl to HPLC-determined Chl indicate that a correction factor of 2 is reasonable for the Mediterranean basin, as now clarified in the revised version of the manuscript: "*The Mediterranean Sea is known to show very small regional variations of the fluorescence-to-Chl ratio (Taillandier et al. 2018), with a mean value close to 2 (1.66±0.28 and 1.72±0.23 for the Western and Eastern Mediterranean, respectively; Roesler et al. 2017)*" (Sect. 2 l. 243–250).

As suggested by the Reviewer, one option would have been to calibrate the float fluorescence measurements based on coinciding HPLC determinations collected during the BOUSSOLE and PEACETIME cruises. Below we show the fluorescence-derived Chl measured at deployment of the floats in the Ligurian (fLig Chl) and Ionian (fIon Chl) Sea, plotted against the corresponding HPLC-determined Chl (Fig. R1). The results displayed in Fig. R1 indicate that the slope factor is not constant with depth, with larger values of the fluorescence Chl to HPLC Chl ratio at depth than within the upper layer of the water column. This trend is especially marked for fLig. In addition, it is at this stage not possible to account for natural variations in the fluorescence-to-chlorophyll ratio that occur both with depth and over the lifetime of the floats, and induce changes in the correction factor. Eventually, in our study, the float fluorescence-based chlorophyll data are used to provide a general context but do not impact the production estimates.

Therefore, we suggest that applying the standard BGC-Argo procedure is more straightforward and equally relevant as applying a custom correction factor. If agreed by the Reviewer and Editor, we will keep the standard factor of 2.

[Figure]

Figure R1: Scatterplot of the chlorophyll concentration (Chl) derived from fluorescence measured at deployment by the BGC-Argo floats in the Ligurian Sea, fLig (left) and Ionian Sea, fIon (right) against corresponding Chl determined by HPLC, within the layer comprised between the surface and ~twice the euphotic layer depth (i.e. 0–100 m for fLig and 0–200 m for fIon). The plain and dashed lines indicate the linear regression model and the slope factor of 2, respectively. The regression model equations are, for fLig and fIon, respectively: fLig Chl = 2.6 HPLC Chl − 0.02 ($r^2$ = 0.73, p < 0.001) and fIon Chl = 1.8 HPLC Chl − 0.02 ($r^2$ = 0.82, p < 0.001).

Lines 242-247:

the difference of float cycle (noon-noon vs sunrise-sunrise) between the two sites might be a problem for the comparative interpretation of the data? Since Lig cycle was run every four days, while Lion did every day, one solution would be to use the cycle noon-noon also at Lion to be similar to Lig. Did the authors try to modify the cycle to look at the potential differences in using another starting point?

Thanks to the Reviewers' comments, we noticed that the sampling scheme of the Ligurian float (fLig) was incorrectly presented in Fig. 2. The fLig float actually measures bio-optical properties from sunrise, noon, sunset and sunrise the following day. In the revised version of the manuscript, we have made appropriate corrections to Fig. 2 (see new version below) and modified the text as follows: "*The fLig float cycle commences with the first profile at sunrise ($t_{sr}$), a second at solar noon ($t_n$), a third profile at sunset the same day ($t_{ss}$), and a fourth profile at sunrise the next day ($t_{sr+1}$). The fLig float then acquires a profile at solar noon 4 days later ($t_{n+4}$), and then restarts 3 days later the acquisition of 4 profiles in 24 hours from sunrise ($t_{sr+7}$).*" (Sect. 2.2 l. 295–299).

Lines 267-277: "daily solar cycle"

268: "Abundance of microorganisms". Which kind of microorganisms is referred to?: phytoplankton, bacteria? Do zooplankton affect the cp or bbp as well?

Yes, we do refer to phytoplankton and heterotrophic bacteria. We have modified our statement as follows: "*The daily solar cycle is a major driver of biological activity in all oceanic euphotic zones, which influences the abundance of microorganisms, including phytoplankton (Jacquet et al. 1998; Vaulot & Marie 1999; Brunet et al. 2007) and heterotrophic bacteria (Oubelkheir & Sciandra 2008; Claustre et al. 2008) and, therefore, the magnitude of the $c_p$ and $b_{bp}$ coefficients*" (Sect. 2.4 l. 343–347).

line 272: cell division. Since cell division is strongly dependent on the species and on the physiology of the species which is modulated by diel light cycle, I suggest adding some information on the physiological changes (e.g., photoacclimation) together with abundances as key-factors determining the success in term of production/division of a community (e.g., Doney, et al (1995) Photochemistry, mixing and diurnal cycles in the upper ocean. J. Mar. Res., 53; Litaker, R. W., et al. (2002) Effect of diel and interday variations in light on the cell division pattern and in situ growth rates of the bloom-forming dinoflagellate Heterocapsa triquetra. Mar. Ecol. Prog. Ser., 232, 63–74; Brunet C. et al. 2008. Phytoplankton diel and vertical variability in photobiological responses at a coastal station in the Mediterranean Sea. J. Plankton Res., 30: 645-654).

In response to the Reviewer's comment, we have first modified the text in order to clarify the different sources of the diel variations in the $c_p$ and $b_{bp}$ coefficients: "*The diurnal increase in $c_p$ or $b_{bp}$ has been primarily attributed to photosynthetic cellular organic carbon production (Siegel et al. 1998), that will first result in an increase in cell size, or an increase in cell abundance and a decrease in cell size following cell division often occurring at night. In addition, the diurnal increase in $c_p$ or $b_{bp}$ may also be caused by variations in cellular shape and refractive index that accompany intracellular carbon accumulation (Stramski & Reynolds 1993; Durand & Olson 1996; Claustre et al. 2002; Durand et al. 2002). The nighttime decrease in $c_p$ or $b_{bp}$ may be explained by a decrease in cellular abundance due to aggregation, sinking or grazing (Cullen et al. 1992), a reduction in cell size and/or refractive index associated with cell division and respiration, the latter involving changes in intracellular carbon concentration with effect on the refractive index (Stramski & Reynolds 1993).*" (Sect. 2.4 l.347–359)

Second, we have mentioned the influence of species and photoacclimation: "*Community composition and cell physiology (in response to diel fluctuations of the light field) might also influence the optical diel variability through their effects on cell size and refractive index. Diel variation in photoacclimation can be important in coastal communities dominated by microplankton (Litaker et al. 2002; Brunet et al. 2008). However, previous studies conducted in*

*oligotrophic environments suggest that photosynthetic growth is the major driver of the diurnal changes in $c_p$ or $b_{bp}$ (Gernez et al. 2002; Claustre et al. 2008). In addition, Claustre et al. (2002), in an experimental work based on Phrochlorococcus, a frequent taxon in oligotrophic regions, show that although non-negligible, the diel variability in photoacclimation is much less pronounced than that in phytoplankton growth"* (Sect. 2.4 l. 359–387).

Line 277: what is the link between microorganism respiration and particle size/refractive index?

Nighttime decrease in refractive index is induced by changes in the intracellular carbon concentration that accompany respiration (e.g. Stramski & Reynolds 1993). In an attempt to clarify this point, we have reworded as follows: *"a reduction in cell size and/or refractive index associated with cell division and respiration, the latter involving changes in intracellular carbon concentration with effect on the refractive index (Stramski & Reynolds 1993)"* (Sect. 2.4 l. 356–359)

Line 308: what does "quasi 1-D framework" mean? It seems to me 1-D?
We have corrected the sentence as suggested by the Reviewer: *"As in previous … we assume a 1D framework"* (l. 418).

Line 325: "ZSCM the depth of the SCM": does the ZSCM refer to the maximum chla fluo depth?

Yes, as this is based on float data as specified in the revised version of the manuscript: *"a Gaussian model is fit to each Chl vertical profile measured by the floats in order to determine the depth interval of the full width half maximum of the SCM"* (l. 288–289).

RESULTS & DISCUSSION
Line 382: remove "in this section"
Done (l. 500).

Line 402: "…the Ionian Sea SCM is located twice as deep (97 m) and is uncoupled from any cp and bbp maxima that occur at shallower depth."  Did the authors have an explanation for this spatial uncoupling between cp and scm?  And why was the cause of the cp maximum in the surface layer?

The contrasted characteristics of the SCM and underlying mechanisms have been reported in previous studies. In response to the Reviewer's comment, we have discussed previous results in support of our present observations. Thus, the paragraph has been completed as follows: *"Hence,*

*the selected regions are representative of two contrasted SCM systems … Such a contrast in the SCM characteristics in relation with the trophic gradient of the environment has already been observed in the Mediterranean Sea (e.g. Lavigne et al. 2015; Barbieux et al. 2019) and on a global scale (e.g. Cullen 2015 and references therein; Mignot et al. 2014; Cornec et al. 2021). These studies report that the depth of the SCM is inversely correlated with the surface Chl (an index of the trophic status) and light attenuation within the water column. Previous studies (Mignot et al. 2014; Barbieux et al. 2019; Cornec et al. 2021) indicate that moderately oligotrophic, temperate conditions are generally associated with a relatively shallow SCM coupled to a maximum in $c_p$ or $b_{bp}$, reflecting an increase in phytoplankton carbon biomass (SBM). In contrast, in the most oligotrophic environments, the vertical distribution of Chl shows a maximum at greater depths and is decoupled from the $c_p$ or $b_{bp}$ vertical distribution. Furthermore, Barbieux et al. (2019) show that, in the northwestern Mediterranean region, the SCM mirrors a biomass maximum located slightly above $Z_{eu}$, which benefits from an adequate light-nutrient regime thanks to a deep winter convective mixing allowing nutrient replenishment in the upper ocean. In the Ionian Sea where the MLD and nutricline are permanently decoupled, the SCM establishes below $Z_{eu}$ as phytoplankton organisms attempt to reach nutrient resources. Prevailing low-light conditions lead to pronounced photoadaptation of phytoplankton. Thus, consistently with previous work, the present observations indicate that the Ligurian Sea SCM is a phytoplankton carbon biomass (SBM) likely resulting from favorable light and nutrient conditions, whereas the Ionian SCM would be essentially induced by photoacclimation of phytoplankton cells.*” (Sect. 3.1 l. 526–550).

I am also wondering if the chl.a maximum at 100 m depth might be called DCM rather than SCM ?

In an analogous manner to Cullen et al. (2015) in their review paper (“*The terms deep chlorophyll maximum and chlorophyll maximum layer are commonly used descriptors that are useful in searches of the literature, but because not all subsurface layers are deep and the development of layers at the sea-surface interface, (e.g., Cullen et al. 1989) is beyond the scope of this review, I use the term SCML here*”), we prefer to use the term “Subsurface Chlorophyll Maximum (SCM)” for both the Ligurian and Ionian systems. If we use DCM (Deep Chlorophyll Maximum) instead, then it will not be relevant to the Ligurian SCM which is not located at great depth.

Line 407: “whereas the Ionian SCM is induced by photoacclimation of phytoplankton cells.” What does it mean? Is there a chl.a maximum? Did the authors suggest that this chl.a maximum is not due to accumulation of microalgae but to an intracellular increase of chl.a due to low light? Can it be a mix of the two (increase in phytoplankton as well as in chla per cell)?

As indicated in the Introduction, and confirmed by the examination of Fig. 3, the SCM in the Ionian Sea appears to reflect an increase in Chl independent of POC (as deduced from the $c_p$ and $b_{bp}$

optical proxies). This does not preclude production at depth, as developed in the Sect. 3.5 below where we estimate a moderate, yet non-negligible, contribution of the SCM layer to community production. We have moderated our sentence accordingly: "would be essentially induced by photoacclimation of phytoplankton cells" (l. 550).

Moreover, the difference between the two systems might be attributed to the fact that SCM depth was 40 m deep in one case and almost 100 m deep in the other. One belongs to the euphotic depth, while the other does not. Microalgal communities seem to be significantly different. All those environmental/ecological properties might explain the low productivity rate in the Ionian sea, compared to the highly productive Ligurian sea. Indeed, other studies (Combet et al., 2011, biogeosciences) reported an increase of POC at depth in the Ionian sea in correspondence of chl.a maximum in summer.

We agree with the Reviewer that the two studied SCM systems are contrasted in their characteristics and underpinning mechanisms, as observed in previous studies and confirmed with the present results. The vertical distribution of POC (as indicated by the $c_p$ and $b_{bp}$ optical proxies) appears to covary with that of Chl in the Ligurian Sea, a feature we do not observe in the Ionian. Our results are quite consistent with those of Crombet et al. (2011) who report, for the oligotrophic period of year 1999, the occurrence of a POC maximum in the upper part of the water column coinciding with the SCM in the Ligurian region, not in the Ionian region. For the oligotrophic period of year 2008, as mentioned by the Reviewer, Crombet et al. (2011) indicate the presence of a second POC maximum at ~100 m that appears to be coupled with the SCM. We attempt to clarify this point in our response to the Reviewer's previous comment (revisions made in l. 528–548).

More generally, I think that light data/information is lacking sometimes. Light is the one of the main driver of primary production as well as of the diel variations of microalgal communities. I suggest adding a figure showing characteristic light profiles. the fig. 10 "only" reports the daily integrated light at the SCM in both sites.

In order to accommodate the Reviewer's suggestion, we have added to Fig. 3 the PAR time series within the water column for both study regions (please see new figure below) and added description of the PAR data in the revised version of the text: "*Consistently, the instantaneous midday PAR values are much lower in the upper layer of the Ligurian Sea (93±70 μE m-2 s-1) than in the Ionian Sea (500±60 μE m-2 s-1) and shows a much more rapid decrease within the water column as phytoplankton biomass absorbs light*" (Sect. 3.1 l. 511–514).

Figure 4: I think that there is a mistake on the colors: chl.a is higher in surface than in scm?

Thank you very much. There was indeed an error in the key. The colors for the surface and SCM layers were reversed. This has been corrected.

Figure 8 is not very informative like this. Please, change the type of figures, we do not really appreciate the potential changes in chla in the three size classes. Also because the right panels report the time variations for only four days.

We believe Fig. 8 is informative because it compares the percent contribution of the three phytoplankton classes on an identical (color) scale in both the Ligurian (BOUSSOLE) and Ionian PEACETIME) regions. Nevertheless, we understand the Reviewer's criticism and hope to address it by providing a new complementary figure as appendix, which displays the vertical distribution of the phytoplankton classes in both study regions. We have also modified the text as follows: "*No marked shift in the community composition is observed during the timeseries, although occasional increase in the contribution of microphytoplankton is observed in the SCM layer, with no clear temporal trend (Fig. 8a and Appendix B)*" (l. 728–731). The new figure is displayed at the end of this document.

Table 3: some values are negative:
- Did the authors estimate the mean of these two parameters mixing positive and negative values ? does it really make sense?
- How can be explained the negative value found for $\delta_!\grave{I}\Delta$ of cp in the euphotic layer in the Ligurian sea, in correspondence with the huge variability (2603.1%)? It would reveal the very high spatial variability in this zone, but the $\delta_!\grave{I}\Delta$ of cp was positive for both the surface and SCM layers.
- Another point: from the fig.3, for the Ligurian sea, euphotic depth includes SCM and surface, while for the Ionian sea, SCM is generally below than the euphotic depth. Does it useful to compare the relative daily variations ($\delta_!\grave{I}\Delta$ and $\delta_«\grave{I}\Delta$, respectively) in the diel cycle of cp and bbp in the euphotic depth between the two systems?

We thank the Reviewer for this question. Our main objective here is to evaluate the daily variations in $b_{bp}$ against those in $c_p$ while comparing our values with those of Kheireddine & Antoine (2014) as a reference. We thus simply decided to apply the method of Kheireddine & Antoine (2014) who accounted for all values, including the negative ones, and report negative values of the $\widetilde{m\Delta}$ for $c_p$ and $b_{bp}$ as well.

In response to the Reviewer's comment, we have calculated the $\widetilde{m\Delta}$ and $\widetilde{r\Delta}$ indices without the negative values of $\tilde{\Delta}c_p$ and $\tilde{\Delta}b_{bp}$. Below we report the results in blue, while those from Table 3

(considering both >0 and <0 $\tilde{\Delta}c_p$ and $\tilde{\Delta}b_{bp}$ values) are reported in black. We remark that disregarding <0 values influences the indices values, but does not necessarily reduce the variability in the range of $\tilde{\Delta}c_p$ and $\tilde{\Delta}b_{bp}$ (except for the euphotic layer in the Ligurian, as noticed by the Reviewer). For example, the $\widetilde{r\Delta}$ index for $\tilde{\Delta}c_p$ in the Ligurian Sea varies from 257% to 223% and 195% to 163% in the surface and SCM layers, respectively, depending on the data accounted for in the calculation (all data vs >0 data only).

| | | Surface layer | | SCM layer | | Euphotic layer | |
|---|---|---|---|---|---|---|---|
| | | $\tilde{\Delta}c_p$ | $\tilde{\Delta}b_{bp}$ | $\tilde{\Delta}c_p$ | $\tilde{\Delta}b_{bp}$ | $\tilde{\Delta}c_p$ | $\tilde{\Delta}b_{bp}$ |
| Ligurian Sea | $\widetilde{m\Delta}$ | 12.7 | -2.3 | 14.5 | 3.8 | -134.9 | -1.1 |
| | $\widetilde{r\Delta}$ | 256.7 | 28.5 | 194.8 | 107.8 | 2603.1 | 20.5 |
| | $\widetilde{m\Delta}$ | 42.5 | 5.3 | 47.2 | 20.9 | 45.6 | 2.4 |
| | $\widetilde{r\Delta}$ | 223.9 | 14.7 | 163.4 | 80.2 | 152.5 | 9.8 |
| Ionian Sea | $\widetilde{m\Delta}$ | 0.55 | 0.23 | 1.16 | 0.06 | 0.39 | 0.21 |
| | $\widetilde{r\Delta}$ | 54.4 | 21.2 | 102.4 | 57.3 | 55.9 | 24.9 |
| | $\widetilde{m\Delta}$ | 6.1 | 2.63 | 27.6 | 11.9 | 8.1 | 3.2 |
| | $\widetilde{r\Delta}$ | 36.5 | 11.3 | 137.6 | 44.7 | 30.6 | 8.8 |

Negative $\widetilde{m\Delta}$ values are caused by rare occurrences of a decrease from sunrise ($t_{sr}$) to sunrise the next day ($t_{sr+1}$), instead of an increase as expected in the 1D assumption framework. The large negative $\widetilde{m\Delta}$ value obtained for $c_p$ (-135%) in the euphotic layer of the Ligurian Sea results from an abrupt decrease (from a maximal 0.07 $m^{-1}$ to a close to minimum value ~0.003 $m^{-1}$) which is also reflected in a large $\widetilde{r\Delta}$ value (2603%). We recognize that these statistics are influenced by an outlier and thus not truly representative of the timeseries.

Our choice was not to remove any data but rather to acknowledge such situations ("*occurrences of negative values…indicating that the 1D assumption is occasionally not satisfied in the lower part of the euphotic layer*" l. 704). Yet, we recognize that showing the diel cycle results for the euphotic layer brings more confusion than information, as our intent here is to examine the amplitude of the daily variations of $b_{bp}$ vs $c_p$ in comparison to previous observations at the BOUSSOLE mooring, available for the surface layer only. Hence, as suggested by the Reviewer, we have removed the information for the euphotic layer in Table 3.

Line 435: efficient?

The Reviewer refers to the following phrase "*In particular, phytoplankton make a larger contribution to cp than bbp, in part due to their strong absorption efficient*". We actually meant "*efficiency*" (l. 593). Thank you.

Line 451: "We compare the cp- and bbp based estimates with primary production estimates computed with the model of Morel (1991)." Is it necessary to use the bbp estimate since the lack of correlation with scm for instance? It does not seem to be relied on phytoplankton production.

In Section 3.2.1, which the Reviewer refers to, we attempt to evaluate the performances of the diel cycle-based method using either the $c_p$ or the $b_{bp}$ coefficient. Based on the results presented in this section, we decide to disregard $b_{bp}$-based estimates and focus only on $c_p$-based estimates. Thus, we would like to conserve the $b_{bp}$-derived results for the purpose of this evaluation, and for consistency in the presentation of the Ligurian and Ionian data. Hence, we believe that, at this stage of the manuscript, both $c_p$- and $b_{bp}$-based production estimates should be presented. We hope we responded to the Reviewer's question appropriately.

Line 544: an/a (correct)

Ok "*which is in fact a SBM*" (l. 709).

Line 546: "the SCM reflects photoacclimation…" the SCM is greatly deeper (below the euphotic depth and the mixing layer), Light is strongly lower than in the Ligurian system SCM, maybe temperature is lower, chl.a is less…consequently the production might be lower than in the other system.

Consistently with our response to previous comments by the Reviewer, we agree that the Ligurian and Ionian SCM systems are contrasted which, to our belief, makes their comparison interesting. As indicated above, we have clarified these contrasted characteristics and underlying mechanisms in the revised version of our manuscript: "*Hence, the selected regions are representative of two contrasted SCM systems … the Ionian SCM would be essentially induced by photoacclimation of phytoplankton cells.*" (l. 526–550).

Lines 595-596: no need to present the data in absolute and in %.

We have made appropriate changes (l. 766, l. 771, l. 773, l. 774, l. 775).

Lines 613-614: Please explain

The Reviewer refers to "*This suggests that the POC in the smaller size fractions are more similar in their respective SCM than in their larger size fractions*". We have rephrased this sentence in order to make it clearer: "*The $b_{bp}$/Chl ratio being more sensitive to small-sized particles than the $c_p$/Chl*

*ratio, these results suggest that, in the SCM layer, the POC in the small size fractions of the Ligurian and Ionian Seas is more similar than that in the large size fractions.*" (l. 788–791).

Line 633: "…increase their intracellular Chl." And/or the fluo/chl.a ratio in relation with the little (or absence) of high-light-induced chl.a quenching

We thank the Reviewer for this suggestion which we have added to our discussion: "*These low values may reflect photoacclimation, by which phytoplankton organisms increase their intracellular Chl, and/or an increase in the fluorescence-to-Chl ratio in relation to limited or null non-photochemical chlorophyll fluorescence quenching.*" (l. 818–819).

CONCLUSIONS
To me, this section is too long, and is like a discussion. Many points raised by the authors in this section were already presented/discussed. I suggest reducing the length of the conclusions.
In order to make the conclusion more synthetic, as recommended by the Reviewer, we have kept only the essential results and the elements of recommendations or perspectives. This reduced the conclusion by about a third of its initial length.

[Figure]

[Figure]

Figure 2: Schematic representation of the diel variations of the depth-integrated bio-optical properties converted to POC biomass (*B*) and the sampling strategies employed in the (a) Ligurian Sea and (b) Ionian Sea. The diamond-shaped symbols indicate schematically the float profile times, labeled with time stamps associated with sunrise (sr), noon (n), sunset (ss) and midnight (m), with the corresponding POC biomass estimated within the considered layer (e.g., $B(t_{sr})$, etc.). The numeric subscripts (+1, +2, +4, +7 or +8) indicate the number of days since the first profile of the summertime time series.

[Figure]

**Figure 3**: Time series of the vertical distribution of the *Chl* (a and d), $b_{bp}$ (b and e), $c_p$ (d and f), and instantaneous midday PAR (d and h), in the Ligurian Sea (left) and the Ionian Sea (right). The euphotic depth ($Z_{eu}$; white line), the Mixed Layer Depth (MLD; black line), the depth of the SCM (magenta line), and the depth of the isopycnal 28.85 expressed as $\sigma_t$ (blue line), are superimposed onto the bio-optical timeseries. The dashed lines indicate the dates at which the $c_p$ and the $b_{bp}$ values in the SCM layer reach a minimum.

[Figure]

**Figure B1**: Vertical distribution of the chlorophyll *a* concentration of the micro- (green), nano- (red) and picophytoplankton (blue) derived from HPLC pigment determinations in the Ligurian Sea (BOUSSOLE site; a–h) and the Ionian Sea (PEACETIME cruise; i). For the Ionian Sea the solid line shows the mean value and the shaded area the standard deviation, calculated over a 4-day window (May 25–28, 2017).